# RANa: Retrieval-Augmented Navigation

**Gianluca Monaci, Rafael S. Rezende, Romain Deffayet, Gabriela Csurka, Guillaume Bono, Hervé Déjean, Stéphane Clinchant and Christian Wolf**
*All authors are affiliated with Naver Labs Europe*    *firstname.lastname@naverlabs.com*
*https://europe.naverlabs.com/research/publications/rana*

Reviewed on OpenReview: *https://openreview.net/forum?id=OWCJ5JfsRB*

## Abstract

Methods for navigation based on large-scale learning typically treat each episode as a new problem, where the agent is spawned with a clean memory in an unknown environment. While these generalization capabilities to an unknown environment are extremely important, we claim that, in a realistic setting, an agent should have the capacity of exploiting information collected during earlier robot operations. We address this by introducing a new retrieval-augmented agent, trained with RL, capable of querying a database collected from previous episodes in the same environment and learning how to integrate this additional context information. We introduce a unique agent architecture for the general navigation task, evaluated on ImageNav, Instance-ImageNav and ObjectNav. Our retrieval and context encoding methods are data-driven and employ vision foundation models (FM) for both semantic and geometric understanding. We propose new benchmarks for these settings and we show that retrieval allows zero-shot transfer across tasks and environments while significantly improving performance.

## 1 Introduction

A realistic setting for robot navigation is the "*unboxing*" scenario, where one robot (or a fleet) is placed in its target setting, started-up, and navigates "*out of the box*". The main appeal of this scenario is the lack of dependency on environment preparation, like scanning the environment, installing external localization systems or generating a floor plan prior to navigation. Most modern methods tackling this scenario, *e.g.* Wijmans et al. (2019); Chaplot et al. (2020a); Yadav et al. (2023a); Bono et al. (2024a), use learning-based approaches in an *episodic setting*: *every* episode in the robot's life is treated as if it was its first. This is obviously sub-optimal: during operation the scene is explored, objects and their locations are observed, affordances found, failure cases encountered etc. Exploiting this information is crucial for designing efficient navigation agents that can continuously improve.

Benchmarks like the *k-item scenario* (Beeching et al., 2020b), *multi-object navigation* (Wani et al., 2020) or *GOAT benchmark* (Khanna et al., 2024) test the capacity of an agent to retain information from previous (sub-)goals and trajectories. Structured latent memory trained end-to-end with RL performs best in these benchmarks (Marza et al., 2022). However, these architectures have only been tested on short (multi-)episodes of 500 to $2,500$ agent steps and are hardly suitable for a real continuous operation: transformer-based agents with self-attention over time suffer from limited context length and the quadratic complexity of attention, while recurrent models are limited by the size of their latent memory and the network capacity growing quadratically with representation size (Jose et al., 2018). Alternative ways to integrate information from initial rollouts are topological maps (Savinov et al., 2018; Sridhar et al., 2024), which require the estimation and generation of a structured graph model from a sequence of posed observations. The extension of these methods to multi-robot scenarios is anything but trivial.

In this work we propose a simple data-driven approach to extend a state-of-the-art navigation agent (Bono et al., 2024a) working out-of-the-box, with capabilities to store visual observations in a global indexed

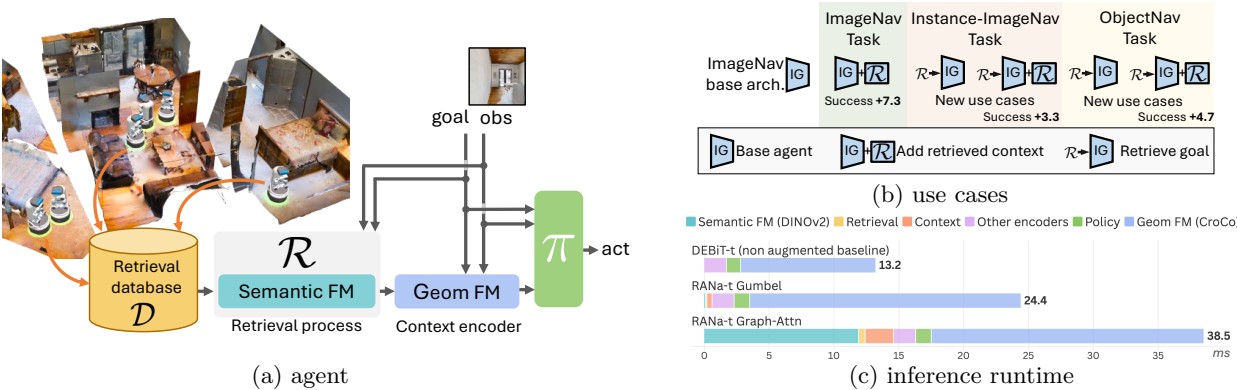

Figure 1: **Retrieval-Augmented Navigation (RANa)**. (a) We augment a navigation agent with context information retrieved from a scene-specific retrieval database, fed with data by one or potentially multiple robots. We leverage *semantic or multi-modal* foundation models for retrieval, and *geometric* foundation models for context encoding. (b) We tackle *ImageNav*, *Instance-ImageNav* and *ObjectNav* tasks, demonstrating performance improvements (*Success +x*), opening *new use cases* from existing models, and zero-shot applications with retrieval only added at test time (ℛ◄[IG]). (c) Retrieval is efficient, negligible compared to vision models, with inference run-time in the same ballpark as non-augmented architectures like DEBiT.

database, share them with a fleet, and retrieve and process them to improve performance during operation. The proposed architecture is general and can be used for different navigation tasks — we evaluate our work on *ImageNav*, *Instance-ImageNav* and *ObjectNav*. While the goal may be specified either as an image or a semantic category, depending on the task, the retrieved information is always visual first person views (FPVs). Data-driven retrieval mechanisms leverage vision foundation models (FM) like DINOv2 (Oquab et al., 2024) to retrieve candidate views, while a geometric FM, CroCo (Weinzaepfel et al., 2022), integrates the retrieved context and provides directional information to the agent, cf. Fig. 1a. Retrieval is also fast and keeps inference time in the same ball park of alternative baselines, cf. Fig. 1c.

This approach is scalable, as the dataset can potentially be very large and can be queried by any number of agents with optimized sub-linear time algorithms. Retrieving from an image database is also considerably simpler and more flexible than using involved memory structures such as metric or topological maps. While we can use any suitable vision or multi-modal FM to represent, organize and retrieve images from an unstructured database, the latter approaches require pose to register observations and synchronization to enable multi-agent data collection: unless pose estimates are absolute (which requires scene preparation), fusing new updates from multiple agents requires a jointly maintained map and distributed algorithms.

We propose the following contributions: (i) new navigation tasks, where the environment is augmented with a retrieval database fed by one or multiple robots; (ii) a unified end-to-end architecture for Retrieval-Augmented Navigation, **RANa**, for multiple tasks; (iii) data-driven retrieval mechanisms that do not require any meta-data beyond raw FPVs; (iv) mechanisms that leverage retrieved information either for additional context, or to replace the goal on the fly, allowing for zero-shot applications, cf. Fig. 1.

## 2 Related Work

**Visual navigation** – The task of navigation has been addressed in robotics research using mapping and planning (Thrun et al., 2005; Macenski et al., 2020), which requires solutions for mapping and localization (Bresson et al., 2017; Mur-Artal & Tardós, 2017; Labbé & Michaud, 2019), planning (Sethian, 1996; Konolige, 2000) and low-level control (Fox et al., 1997; Rösmann et al., 2015). These methods depend on accurate sensor models, filtering, dynamical models and optimization schemes. In contrast, end-to-end models learn deep representations such as flat recurrent states (Yadav et al., 2023a; Bono et al., 2024a), occupancy maps (Chaplot et al., 2020b), semantic maps (Chaplot et al., 2020a), latent metric maps (Henriques & Vedaldi, 2018; Parisotto & Salakhutdinov, 2018; Beeching et al., 2020b; Marza et al., 2022), topological maps (Savinov et al., 2018; Beeching et al., 2020a; Chaplot et al., 2020c; Sridhar et al., 2024), self-attention (Fang et al., 2019;

Du et al., 2021; Chen et al., 2022; Reed et al., 2022; Zeng et al., 2025) or implicit representations (Marza et al., 2023; Kwon et al., 2023). Then, they map these representations into actions, using RL (Jaderberg et al., 2017; Mirowski et al., 2017; Bono et al., 2024c), imitation learning (Ding et al., 2019) or by maximizing navigability (Bono et al., 2024b).

In this work, our goal is to address end-to-end learning of representations and policies with RL. Although this problem is nearly solved for *PointNav*, where the agent must navigate to a relative GPS coordinate (Savva et al., 2019; Wijmans et al., 2019), it remains a challenge for other goal specifications.

Here we focus on *ImageNav*, where an image is provided as goal, requiring geometric understanding of the scene. A common variant is *Instance-ImageNav* (Krantz et al., 2023), where the goal image depicts a specific object in the environment and is taken with camera parameters different from those of the agent. Recently, the end-to-end model DEBiT (Bono et al., 2024a) achieved state-of-the-art performance in *ImageNav* leveraging the geometric FM CroCo (Weinzaepfel et al., 2022) to estimate relative poses between goal and agent's observation. In this work, we build upon DEBiT and augment it with retrieval, naturally extending it to multi-robot setting and enabling zero-shot *ObjectNav* and *Instance-ImageNav*. To the best of our knowledge no other method performs all these tasks competently, the closest competitor being ZSON (Majumdar et al., 2022) which tackles *ImageNav* and zero-shot *ObjectNav*, with considerably lower performance (see Section 5).

**Retrieval-augmented control** – The idea of retrieving information from a dataset of past experiences has been introduced in earlier works. Among them, episodic control methods for RL (Blundell et al., 2016; Pritzel et al., 2017; Lin et al., 2018; Hansen et al., 2019; He et al., 2024; Hu et al., 2021) act on successful experiences by re-employing Q-value estimates, enhancing sample efficiency during training. In contrast, we exploit retrieved information from past experiences of a robot fleet to enhance navigation performance and enable zero-shot task generalization. Goyal et al. (2022) augment an RL agent with a retrieval process parameterized as a neural network that has access to a dataset of past trajectories. Here instead, we take inspiration from Humphreys et al. (2022) that, to our knowledge, was the first work to propose a generic non-parameterized retrieval augmentation for an RL agent with previously learned embeddings. Alleviating the need to pre-train embeddings with RL, we use vision foundation models to process stored and retrieved images. This enables zero-shot transfer and allows to adapt the approach to various navigation tasks.

Retrieval-Augmented Planning (Kagaya et al., 2024) uses a memory of successful task executions, comprising plans, actions and observation sequences. Past experiences relevant to the task at hand are leveraged to guide planning with an LLM agent based on task constraints and state similarity. Xie et al. (2024) build a hierarchical topological memory to prompt an LLM for reasoning and goal-finding in a navigable environment. In contrast, we only store images derived from unstructured observations without any associated metadata.

**Navigation supported by prior rollouts** – Other navigation methods exploit previous observations to construct a topological map of the explored scene, and use it to guide navigation (Savinov et al., 2018; Chaplot et al., 2020c; Shah & Levine, 2022; Sridhar et al., 2024). In contrast, our database is unstructured and we can query it to retrieve goal and context images with simple image retrieval. Also, here context images are not used as navigation waypoints, but as recommendations, exploited or not by the agent, leading to navigation performance robust to potentially misleading context, as shown in Section 5.

In Table 1 we compare RANa with selected navigation methods discussed above in the continual navigation scenario. Most approaches require depth and pose and/or more involved (metric or topological) map updates, which need at least to register observations. In contrast, RANa does not require depth or pose, the database update and retrieval is straightforward, and naturally supports multiple data collection robots without the need of synchronization. While SLAM-based solutions (Engel et al., 2014; Mur-Artal & Tardós, 2017) are well established for continual navigation, with RANa we propose a lightweight data-driven solution that can be easily scaled up to multiple robots and can tackle *ImageNav*, *Instance-ImageNav* and *ObjectNav* use cases.

## 3    Retrieval-augmented agent

We target navigation in 3D environments, where an agent is tasked to navigate from a starting location to a goal and receives at each timestep $t$ an image observation $\mathbf{x}_t$. Our method is general and the agent can be trained for a diverse range of tasks, but without loss of generality in this presentation we focus on *ImageNav*,

| Method and representation | Img Nav | Obj Nav | **Not** required: pose | depth | Easiness of representation update | | Multiple collectors[‡] |
|---|---|---|---|---|---|---|---|
| GOAT (Khanna et al., 2024) | ✓ | ✓ | ✗ | ✗ | ∼ | map registration | ✗ |
| 3D Scene graphs (Yin et al., 2024) | ✗ | ✓ | ✗ | ✗ | ∼ graph+map registration | | ✗ |
| Topo maps (Savinov et al., 2018; Shah & Levine, 2022) | ✗ | ✓ | ✗ | ✗ | ∼ | graph update | ✗ |
| Semantic maps (Chaplot et al., 2020a) | ✗ | ✓ | ✗ | ✗ | ∼ | map registration | ✗ |
| Sem. Implicit (Marza et al., 2023) | ✗ | ✓ | ✗ | ✗ | ✗ | ∇ update | ✗ |
| Feature SLAM (Mur-Artal & Tardós, 2017) | ✗ | ✗ | ✗ | ✓ | ∼ | pose graph | ✗ |
| Dense SLAM (Engel et al., 2014) | ✗ | ✗ | ✗ | ✓ | ∼ | map registration | ✗ |
| **RANa** | ✓ | ✓ | ✓ | ✓ | ✓ | store (matrix update) | ✓ |

[‡] *In order to support multiple collectors, a method should either be pose free, like RANa, or use distributed algorithms to synchronize, or absolute pose (and not episodic).*

Table 1: RANa compared to SOTA in continual navigation. RANa is a flexible, data-driven solution that can easily exploit data from previous episodes, in particular collected by multiple robots: it only requires storing images in a dataset without any additional metadata, and pre-compute retrieval features, *e.g.*DINOv2. In case of *dynamic context* (cf. Section 4.2), a similarity matrix, computed offline, is also incrementally updated when new elements are added.

where the agent receives a goal image $\mathbf{g}$. An extension to the *ObjectNav* task is presented in Section 6. The action space is discrete, $\mathcal{A} =\{$MOVE FORWARD 0.25m, TURN LEFT $10°$, TURN RIGHT $10°$, and STOP$\}$. Navigation is considered successful if the STOP action is selected when the agent is within 1m of the goal position in terms of geodesic distance.

Our set-up is based on the *ImageNav* task of the Habitat simulator and platform (Savva et al., 2019). We extend it with a retrieval dataset $\mathcal{D}$ and a retrieval mechanism $\mathcal{R}$, which will be detailed in Section 4. The dataset $\mathcal{D} = \{\mathbf{x}_i^{\mathcal{D}}\}$ contains FPVs indexed by $i$ and stemming from previous episodes or from exploratory rollouts in the scene. While it is in principle possible to store additional meta-data associated to data collection, like approximate pose, performed actions, reward, success or value (in an MDP sense, when available), in Section 5 we show that access to FPVs alone provides rich, useful information to the agent.

The agent queries this dataset $\mathcal{D}$ with the retrieval mechanism $\mathcal{R}$, which can be potentially used at each time step $t$. $\mathcal{R}$ uses the goal $\mathbf{g}$, and optionally the current observation $\mathbf{x}_t$, to retrieve a set of FPVs $\mathcal{R}(\mathbf{x}_t, \mathbf{g}) \longrightarrow \{\mathbf{r}_{t,1}, \mathbf{r}_{t,2}, \ldots \mathbf{r}_{t,N}\}$, that is used as context information to improve navigation. Depending on the use case, the use of observation $\mathbf{x}_t$ in the retrieval process may be optional (as in the *static context* in Section 4) — if not provided, retrieval can be done once per episode, as opposed to every time step.

We propose to integrate the retrieval mechanism $\mathcal{R}$ into a recurrent agent that maps observations and goals to actions. The non-augmented base agent can be described as[1]:

$$\tilde{\mathbf{x}}_t = x(\mathbf{x}_t) \qquad \text{// perceive}$$
$$\tilde{\mathbf{g}}_t = g(\mathbf{x}_t, \mathbf{g}) \qquad \text{// compare obs+goal}$$
$$\mathbf{h}_t = h(\mathbf{h}_{t-1}, \tilde{\mathbf{x}}_t, \tilde{\mathbf{g}}_t, l(\mathbf{a}_{t-1})) \text{ // update recurrent state}$$
$$\mathbf{a}_t \sim \pi(\mathbf{h}_t), \qquad \text{// act}$$

where $x$ and $g$ are trainable encoders, $h$ is the update function of a GRU (Cho et al., 2014) which maintains a hidden state $\mathbf{h}_t$ over time; $l$ is an embedding function, and $\pi$ is the policy. For clarity we have omitted the equations of the gating functions.

Then, the retrieval-augmented agent is defined as:

$$\tilde{\mathbf{x}}_t = x(\mathbf{x}_t) \qquad \text{// perceive}$$
$$\tilde{\mathbf{g}}_t = g(\mathbf{x}_t, \mathbf{g}) \qquad \text{// compare obs+goal}$$
$$\tilde{\mathbf{c}}_t = c(\mathcal{R}(\mathbf{x}_t, \mathbf{g}), \mathbf{x}_t, \mathbf{g}) \qquad \text{// retrieve and encode context}$$
$$\mathbf{h}_t = h(\mathbf{h}_{t-1}, \tilde{\mathbf{x}}_t, \tilde{\mathbf{g}}_t, \tilde{\mathbf{c}}_t, l(\mathbf{a}_{t-1})) \text{ // update recurrent state}$$
$$\mathbf{a}_t \sim \pi(\mathbf{h}_t), \qquad \text{// act}$$

where $c$ is an encoder which compresses the retrieved context into a compact form. It is trained end-to-end together with the agent, as described in Section 3.1, while the retrieval process $\mathcal{R}$, described in Section 4,

---

[1]We denote functions with *italic*, tensors with **bold face**, and encoded tensors with ∼**bold face**.

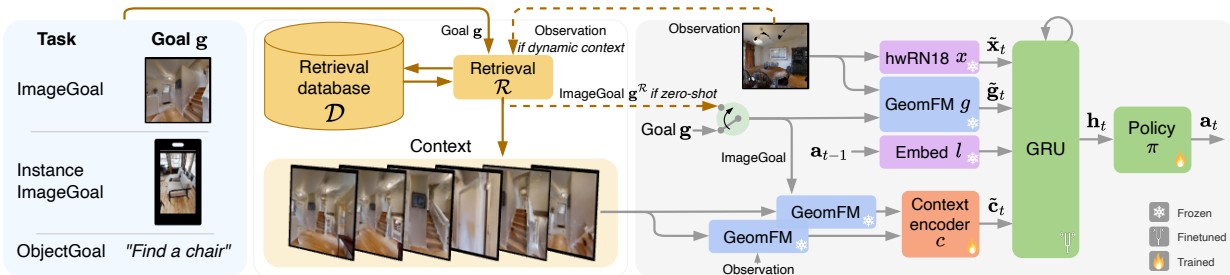

Figure 2: We propose an architecture for **Retrieval-Augmented Navigation architecture − RANa**, that addresses all use cases of Fig. 1b. Dashed arrows are optional depending on the navigation task.

leverages pre-trained models. The observation encoder $x$ is implemented as a half-width ResNet-18 (hwRN18) architecture (He et al., 2016).

The image goal $\mathbf{g}$ is compared to the observation $\mathbf{x}_t$ through a function $g(\mathbf{x}_t, \mathbf{g})$ implemented with a binocular visual encoder from Bono et al. (2024a) that returns an embedding representing information about the goal's direction, pose, and visibility. This encoder was fine-tuned from the foundation model CroCo (Weinzaepfel et al., 2022), solving as pre-text tasks relative-pose and visibility estimation.

Additionally, an image goal $\mathbf{g}^{\mathcal{R}}$ can be retrieved from $\mathcal{D}$ given the task goal $\mathbf{g}$, allowing the *ImageNav* model to address tasks in a zero-shot way. For example, an *ImageNav* agent can perform *ObjectNav* by retrieving a goal image $\mathbf{g}^{\mathcal{R}}$ from the database $\mathcal{D}$ given a goal category ($\mathcal{R}$➤IG, $\mathcal{R}$➤IG➤$\mathcal{R}$). In this case, the goal comparison function $g$ is computed between $\mathbf{x}_t$ and $\mathbf{g}^{\mathcal{R}}$ (instead of the original goal $\mathbf{g}$) as $g(\mathbf{x}_t, \mathbf{g}^{\mathcal{R}})$. We explore this use of retrieval for zero-shot task generalization in Section 5.2.

In summary, the agent receives at each timestep a new visual observation $\mathbf{x}_t$, which it encodes, compares with the goal $\mathbf{g}$, and enriches with additional context information retrieved from the database $\mathcal{D}$. This representation is then passed to a recurrent policy. See Fig. 2 for a visual overview.

### 3.1 Trainable context encoder $c$

At each timestep $t$, the task of the context encoder $c$ is to extract useful information from the list $\mathcal{R}(\mathbf{x}_t, \mathbf{g}) = \{\mathbf{r}_{t,n}\}_{n=1}^N$ of retrieved context images. To do so, we argue that the comparison of retrieved images, goal and current observation is of geometric nature and can be performed by leveraging a pre-trained pose-estimator, or by a geometric FM that could extract relevant directional information to be given to the agent. Therefore, we explore the same geometric FM CroCo used for the binocular encoder $g$ to translate the context $\{\mathbf{r}_{t,n}\}_{n=1}^N$ into a set of embeddings $\{\mathbf{e}_{t,n}\}_{n=1...N}$ by passing them through the binocular encoder along with the current observation and the goal image:

$$\mathbf{e}_{t,n} = [g(\mathbf{x}_t, \mathbf{r}_{t,n}) \ g(\mathbf{g}, \mathbf{r}_{t,n})]. \tag{1}$$

At this point, we obtain a list of embeddings $\mathbf{e}_{t,n}$ that benefit from the rich inductive bias provided by the geometric FM, including information about pose and visibility of the context image $\mathbf{r}_{t,n}$ from the observed image $\mathbf{x}_t$ and goal image $\mathbf{g}$. The list is compressed into a single context embedding $\tilde{\mathbf{c}}_t = c(\mathcal{R}(\mathbf{x}_t, \mathbf{g}), \mathbf{x}_t, \mathbf{g})$ that provides useful information to the agent, and for this we explore two variants.

**Variant 1: Gumbel soft-max selector** – The agent selects a single embedding from the list by predicting a distribution over items in the retrieved list followed by a Gumbel soft-max sampler (Jang et al., 2017):

$$\tilde{\mathbf{c}}_t \sim \sigma(\{\alpha_n\}_{n=1}^N), \quad \alpha_n = \texttt{Linear}(\mathbf{e}_{t,n}) \tag{2}$$

where $\sigma$ is the soft-max distribution over items in the list. We perform the sampling and the computation of gradients using the Gumbel soft-max trick (Jang et al., 2017). We motivate this choice from the pre-training of the geometric FM $g$ in Eq. (1): the latent vector $z = g(a, b)$ has been pre-trained to provide overlap

information between images $a$ and $b$, which we conjecture to be highly correlated with the relevance of the corresponding context item. Additionally, since the weights $\alpha_i$ are calculated independently, the model can be zero-shot applied to any context size at test time.

**Variant 2: Attention-based encoder** – A transformer-based model is potentially capable of integrating multiple context tokens $\{\mathbf{e}_{t,n}\}_{n=1}^N$ into a single fused embedding, effectively changing and enriching the original embedding space. We use a transformer encoder layer with 2 layers of 4-heads self-attention, which receives as input the $N$ context tokens summed with positional embeddings, and predicts the context feature $\tilde{\mathbf{c}}_t$ by concatenating the output tokens and feeding them to a 2-layer `MLP`.

## 4 Retrieval mechanisms for RANa

The retrieval process performs a nearest neighbor search on image representations obtained with DI-NOv2 (Oquab et al., 2024) resulting in a retrieved context. We investigate two context building strategies:

1. *Static context*, where only the goal is used to build a fixed context kept throughout the episode,
2. *Dynamic context*, which also depends on the current observation and is recomputed at each timestep.

### 4.1 Static context

We build the static context by ranking images from the dataset based on DINOv2 feature similarity with the goal and selecting the top-$N$ nearest neighbors.

**Diversity** – providing multiple near-duplicates to the selector might not be very helpful. We therefore resort to *Maximum Margin Relevance* (MMR) from Carbonell & Goldstein (1998) re-ranking to increase diversity. Assuming a relevance score vector $\omega$ (DINOv2 similarities) providing a ranked shortlist (top-100 in our experiments), and a matrix $\Omega$ encoding the similarity for each pair of images in the ranked shortlist, MMR greedily selects at each step (rank) $i$ the element $r_i$ that maximizes the re-ranking criterion:

$$\text{MMR}(r_i) = \beta\omega(r_i) - \left(1 - \beta \max_{r_j \in \mathcal{S}_{i-1}} \Omega(r_i, r_j)\right),$$

where $\beta \in [0,1]$ is a mixture parameter (set to 0.5 in all our experiments) and $\mathcal{S}_{i-1}$ is the set of images already selected. We then select the top-$N$ highest MMR. The impact of MMR is studied in Appendix D.

### 4.2 Dynamic context set

Dynamic context exploits the current observation at each time step, and we leverage this to use the context set to provide intermediate waypoint images to the augmented agent. Similar to topological maps we do this by building a graph and calculating shortest paths. However, in stark contrast to classical topological maps, (i) the graph is built from the images of the retrieval dataset only and can be pre-computed, (ii) no pose information is required, we use visual similarities only, and (iii) waypoint images are recommendations only, exploited or not by the agent through its retrieval context.

**DINOv2-Graph** – Given $\mathcal{D}$, we build the affinity matrix $\Omega_{\mathcal{D}}$ containing the similarity between all pairs of images in $\mathcal{D}$ using DINOv2 feature cosine similarity (Oquab et al., 2024). From $\Omega_{\mathcal{D}}$ we can derive a graph $\mathcal{G} = \{\mathcal{N}, \mathcal{E}\}$, whose nodes $\mathcal{N}$ are images $\mathbf{x}_i^{\mathcal{D}} \in \mathcal{D}$ and edges $\mathcal{E}$ are weighted by similarity. Then, we find in $\mathcal{D}$ the most similar images $r_1^{\mathbf{x}_t}$ and $r_1^{\mathbf{g}}$ to the observation $\mathbf{x}_t$ and goal $\mathbf{g}$, respectively. Finally, we populate the context $\mathcal{R}(\mathbf{x}_t, \mathbf{g})$ with $r_1^{\mathbf{x}_t}$, $r_1^{\mathbf{g}}$ and images sampled on the shortest path in the graph between them.

Fig. 3 shows three examples of shortest paths on the DINOv2-Graph, in green. For comparison only, we also visualize the shortest paths obtained by the ground-truth (GT) "Pose Graph", not available to the agent. The path on the DINOv2-Graph well approximates the GT path. We provide more details about the graph construction and its relation to topological maps in Appendix C.

**Retrieving the goal** – independent of the creation of a retrieval context, static or dynamic, in zero-shot settings we also use retrieval to replace goal $\mathbf{g}$ with a retrieved image goal $\mathbf{g}^{\mathcal{R}}$ ($_{\mathcal{R}}$➤$_{\text{IG}}$, $_{\mathcal{R}}$➤$_{\text{IG}}$$_{\boxed{\mathcal{R}}}$), setting $\mathbf{g}^{\mathcal{R}}$ as the nearest neighbor of $\mathbf{g}$ from the dataset $\mathcal{D}$.

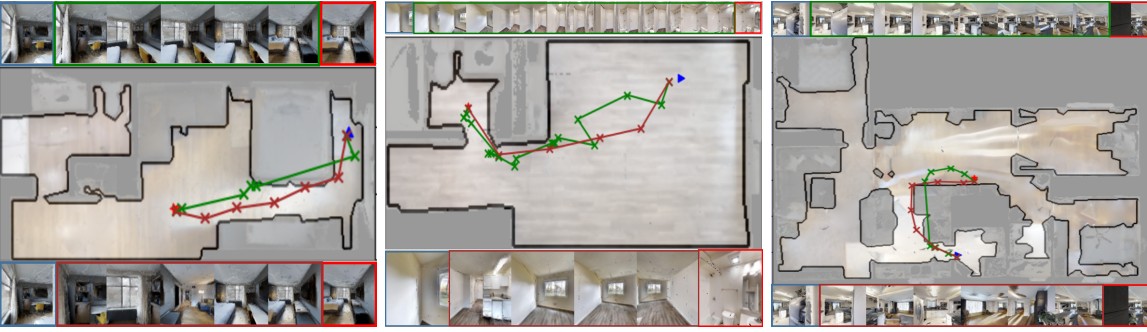

Figure 3: **Shortest paths on DINOv2-Graph and Pose Graph** - The most similar image $r_1^{\mathbf{x}_t} \in \mathcal{D}$ to the goal is highlighted in red and localized with a red star on the floorplan, while $r_1^{\mathbf{g}} \in \mathcal{D}$, in blue, is the most similar image to the current observation. Green crosses connects images along the shortest-path in the DINOv2-Graph (corresponding images in green) and purple crosses refer to poses on the Pose Graph, built using distances between GT camera poses, with corresponding images highlighted in purple at the bottom of the map.

## 5 Experimental results

We train and evaluate our agents on the Habitat simulator and platform (Savva et al., 2019) according to the standard *ImageNav*, *Instance-ImageNav* and *ObjectNav* task definitions.

**Retrieval-augmented navigation benchmarks** – There is no publicly available benchmark that provides a database of observations that can be exploited during training and evaluation. We propose to derive our benchmarks from standard Habitat tasks (Savva et al., 2019), modifying them as little as possible to enable training and evaluating retrieval-augmented agents. We use the Gibson dataset (Xia et al., 2018), consisting of 72 train and 14 eval scenes, and for each scene, unless otherwise stated, we generate a retrieval set $\mathcal{D}$ of *as little as* $1,000$ *FPV images* by letting agents navigate the environment and store images in $\mathcal{D}$. During training and evaluation, the retrieval-augmented agent can select dataset images from the scene it operates in. Additional details on dataset creation are provided in Appendix A].

**Implementation details** – We base our agents on the DEBiT architecture by starting from the official codebase and weights provided by the authors[2], and extend them as described in Section 3. We consider two DEBiT backbones, *base* (DEBiT-b) with a binocular encoder with 55M parameters, and *tiny* (DEBiT-t), with a 17M-parameters encoder, to allow faster execution of ablation studies. The corresponding retrieval-augmented models are RANa-b and RANa-t. In Section 5.2 we use the same agents and weights in zero-shot fashion for *Instance-ImageNav* and *ObjectNav* when the task goal is replaced with a retrieved image goal ($\mathcal{R}$⇢ IG , $\mathcal{R}$⇢ IG ⇢ R ). More details about the agent architecture are provided in Appendix B.

**Model training** – All models, retrieval-augmented or not, are trained with PPO up to 200M steps. Unless specified otherwise, the geometric FM $g$ and encoders $x$ and $l$ are loaded from DEBiT and kept frozen, as is DINOv2. For retrieval-augmented agents, the context encoder $c$ and policy network $\pi$ are learned from scratch. Compared to DEBiT, the GRU memory of RANa receives the additional context input $\tilde{\mathbf{c}}_t$, therefore we expand the DEBiT GRU input matrix to reach the required dimension and finetune it. Further details and ablations of this choice are reported in Appendix D. The reward definition is inspired by *PointNav* (Chattopadhyay et al., 2021) and *ImageNav* (Bono et al., 2024a) rewards and given as:

$$r_t = \mathrm{K} \cdot \mathbf{1}_{\text{success}} - \Delta_t^{\text{Geo}} - \lambda,$$

where $K{=}10$, $\Delta_t^{\text{Geo}}$ is the increase in geodesic distance to the goal, and slack cost $\lambda{=}0.01$ encourages efficiency.

**Retrieval** – We consider two context encoding approaches, Gumbel and Attention, and two retrieval mechanisms, static based on DINOv2, and dynamic based on DINOv2-Graph. The context size is $N{=}8$ in all experiments (see Appendix D for ablations). For dynamic variants requiring DINOv2 to be applied to each

---

[2]https://github.com/naver/debit

| | Model | Retrieval | Context Enc. | SR | SPL |
|---|---|---|---|---|---|
| IG | DEBiT-t (Bono et al., 2024a) | - | - | 79.9 | 51.4 |
| IG+R | **RANa-t** *static* | DINOv2 | Gumbel | 83.0 | 52.3 |
| IG+R | **RANa-t** | DINOv2 | Attention | 81.9 | 54.3 |
| IG+R | **RANa-t** | DINOv2-Graph | Gumbel | 82.0 | 51.5 |
| IG+R | **RANa-t** *dynamic* | DINOv2-Graph | Attention | **84.7** | **63.7** |
| IG | DEBiT-b (Bono et al., 2024a) | - | - | 83.4 | 56.8 |
| IG+R | **RANa-b** *static* | DINOv2 | Gumbel | 88.3 | 57.6 |
| IG+R | **RANa-b** | DINOv2 | Attention | 84.3 | 58.3 |
| IG+R | **RANa-b** | DINOv2-Graph | Gumbel | 89.7 | 60.4 |
| IG+R | **RANa-b** *dynamic* | DINOv2-Graph | Attention | **90.7** | **71.8** |

Table 2: **Effect of retrieving context**, RANa-t and -b models.

| Model | Odom | SR | SPL | Binocular encoder |
|---|---|---|---|---|
| OVRL (Yadav et al., 2023b) | ✓ | 54.2 | 27.0 | Finetuned |
| VC1-ViT-L (Majumdar et al., 2023) | ✓ | 81.6 | – | Finetuned |
| OVRL-v2 (Yadav et al., 2023a) | ✓ | 82.0 | 58.7 | Finetuned |
| ZSEL (Al-Halah et al., 2022) | ✗ | 29.2 | 21.6 | Obs. & policy frozen, goal from scratch |
| ZSON (Majumdar et al., 2022) | ✗ | 36.9 | 28.0 | Observation finetuned, goal frozen |
| *FGPrompt (Sun et al., 2024a)** | ✗ | *92.3* | *68.5* | *Trained from scratch* |
| DEBiT-b no adapters (Bono et al., 2024a) | ✗ | 83.0 | 55.6 | Frozen |
| **RANa-b** *dynamic* | ✗ | 90.7 | 71.8 | Frozen |
| DEBiT-L w. adapters (Bono et al., 2024a) | ✗ | 94.0 | 71.7 | Finetuned with adapters |
| **RANa-L w. adapters** *dynamic* | ✗ | **94.5** | **76.2** | Finetuned with adapters |

Table 3: *ImageNav* **state-of-the-art results.** *this work sets the Habitat simulation parameter `sliding=True`, simplifying the task significantly and making it incomparable with other methods (cf. (Monaci et al., 2025)).

observation, we again leverage retrieval to speed up RL training significantly: during training, we replace observations $\mathbf{x}_t$ by their nearest dataset image, allowing to use pre-computed DINOv2 features.

**Metrics** – Navigation performance is evaluated by success rate (SR), *i.e.*, the percentage of episodes terminated within a distance of <1m from the goal after the agent has called the `STOP` action, and SPL (Anderson et al., 2018), *i.e.*, SR weighted by the optimality of the path:

$$SPL = \frac{100}{N} \sum_{i=1}^{N} S_i \frac{\ell_i^*}{\max(\ell_i, \ell_i^*)},$$

where $S_i$ is a binary success indicator in episode $i$, $\ell_i$ is the agent path length and $\ell_i^*$ the shortest path length.

### 5.1 Retrieval improves navigation

Table 2 shows results of the proposed RANa agents with retrieved context on the *ImageNav* task, exhibiting significant performance improvements across all settings compared to the DEBiT baselines[3]. For static context retrieval the Gumbel encoder shows better results, arguably because there is no inherent ordering in the context and the retrieved CroCo-based features contain strong information about the relative pose between context elements and observation. To simplify notation, from now on we label the combination of static context retrieval and Gumbel encoding *static*. The attention-based encoder works better for the dynamic graph-based context, where reasoning about the order of context elements becomes important. We label this graph-based context combined with attention encoding *dynamic*. It obtains the best results, with a huge boost in SPL (+15 and +12.3 points) compared to the DEBiT-b and DEBiT-t baselines, respectively.

**Comparison to *ImageNav* SOTA** – In Table 3 we compare RANa with state-of-the-art (SOTA) *ImageNav* models. These typically finetune (with a navigation loss) visual encoders based on CLIP (Majumdar et al.,

---

[3]To make baselines comparable, loaded checkpoints continued training for the same additional 200M steps, updating GRU and policy only.

| Model | Nr Par | SR | SPL | Tot. runtime (ms) | DINOv2 | Retrieval | Context | $l+x$ | GRU+$\pi$ | CroCo |
|---|---|---|---|---|---|---|---|---|---|---|
| DEBiT-t | 27M | 79.3 | 50.0 | 13.2 | - | - | - | 1.7 | 1.1 | 10.4 |
| DEBiT-b | 66M | 83.0 | 55.6 | 25.2 | - | - | - | 1.7 | 1.1 | 22.4 |
| **RANa-t** *static* | 32M | 83.0 | 52.3 | 24.4 | 0.1 | 0.1 | 0.4 | 1.7 | 1.2 | 20.9 |
| **RANa-t** *dynamic* | 34M | **84.7** | **63.7** | 38.5 | 11.9 | 0.5 | 2.2 | 1.7 | 1.2 | 21.0 |

Table 4: **Inference runtime** of different model components, in ms, timed on one Nvidia H100-80G GPU. Retrieval overhead is negligible, most runtime is spent on image feature computations (DINOv2, CroCo), which increase with retrieval because of additional operations needed to select and compare retrieved elements.

| DB size | 100 | 1k | 10k | 20k |
|---|---|---|---|---|
| 1k | 81.7 (*50.1*) | 82.7 (***52.6***) | 82.1 (*51.2*) | 82.8 (*51.9*) |
| 10k | 80.6 (*49.4*) | **83.0** (*52.3*) | 79.8 (*46.3*) | 82.2 (*51.5*) |

Table 5: **Impact of retrieval database size** - SR (*SPL*) for RANa-t with *static* context using different database sizes, combinations of train and test conditions.

2022), ViT (Yadav et al., 2023b; Majumdar et al., 2023; Yadav et al., 2023a) and geometric FMs (Bono et al., 2024a) used for observation-goal comparison. Not only RANa considerably boosts the performance of the strong DEBiT-b model, but it also enhance the SOTA performance of DEBiT-Large (Bono et al., 2024a) finetuned with adapters (Chen et al., 2023). While SR is only marginally improved, arguably because it is already very high (94) and remaining failure cases are not solvable using context, path efficiency (expressed by SPL) increases significantly, from 71.7 to 76.2 points.

**Retrieval is efficient** – Table 4 shows model size and runtime of different components of selected DEBiT and RANa models. Retrieval time is negligible and scales favorably with the dataset size thanks to efficient approximate nearest neighbor search libraries such as Faiss (Douze et al., 2024). Inference time is slightly higher for RANa because the encoder $c$ computes CroCo features between context images and observation and goal (while DEBiT only runs CroCo between observation and goal). Still, RANa-t is smaller and faster than DEBiT-b while achieving similar performance, suggesting that retrieval augmentation can be a frugal alternative to model scaling. The variant with *dynamic* context adds the compute time required to run DINOv2 on each observation, but its runtime remains in the same ballpark of other agents.

**100 dataset images suffice to improve performance** – Table 5 shows that the size of the retrieval database can be small, 100 images per scene suffice for the considered Gibson scenes, and performance plateaus at 1k images. Additional experiments for *Instance-ImageNav* and *ObjectNav* are presented in Appendix D.

**The geometric FM provides essential directional information** – and plays a crucial role in the effectiveness of RANa. We validate this intuition by training a RANa-t model with *static* context where the frozen CroCo encoders $g$ in Eq. (1) are replaced by a hwRN18. This network has the same architecture of the encoder $x$, but takes as input a $9{\times}112{\times}112$ tensor formed by channel-concatenating observation, goal and context, and outputs the context features $\mathbf{e}_{t,n}$ fed to the Gumbel selector. This encoder is trained together with the rest of the RANa model. Table 6 shows that this model achieves 79.1 SR, against 83.0 SR of RANa-t: in this setting, the agent is not capable of exploiting context information and performance remain at DEBiT-t baseline level, validating the importance of the geometric pre-training of CroCo FM in Bono et al. (2024a).

**Context robustness and failure cases** – Table 7 studies the robustness of RANa-t to potentially misleading context elements: (i) in domain: expected retrieved elements, and (ii) random: random images from the scene dataset. Providing random context images results in performance drop of both RANa *static* and *dynamic* to baseline levels. These results support two key properties of the proposed approach: the context brings useful

| | Model | Retrieval Features | SR | SPL |
|---|---|---|---|---|
| IG | DEBiT-t | - | 79.9 | 51.4 |
| IG+R | **RANa-t** *static* | CroCo Geometric FM | **83.0** | **52.3** |
| IG+R | **RANa-t** *static* | hwRN18 | 79.1 | 48.2 |

Table 6: **Impact of Geometric FM** on context representation.

| Model | in domain | random |
|---|---|---|
| **RANa-t** *static* | 83.0 (*52.3*) | 77.0 (*47.1*) |
| **RANa-t** *dynamic* | **84.7** (***63.7***) | 80.3 (*50.2*) |

Table 7: **Impact of retrieved context** - SR (*SPL*) for RANa-t models on different test conditions .

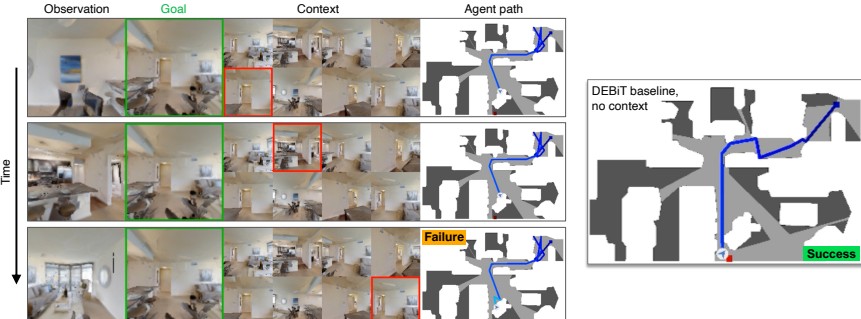

Figure 4: **A failure case**: RANa *static* reaches the zone of the goal image (bottom on the map) and then starts turning around, circling through context items, while the DEBiT baseline (on the right) quickly navigates to the goal.

information and RANa learns to exploit it, and also, RANa can filter out irrelevant context elements and is robust to misleading random retrieval. In fairness, we observe rare occurrences in which access to context deteriorates the performance of the baseline agent. In these cases RANa seems "distracted" by context items, and Fig. 4 shows a typical instance of this phenomenon: the baseline DEBiT agent (trajectory on the right panel) quickly gets to the goal, while RANa reaches the goal area, but starts turning on itself, cycling multiple times through relevant, but "distracting", context images, leading to time-out.

**Training curves** – RANa agents are trained starting form DEBiT checkpoints, training curves are stable and quickly converge to high values. Fig. 5 shows the evolution of Return and Success Rate (SR) during training for RANa-t with *static* Gumbel context encoder, in pink, and DEBiT finetuned for additional 200M steps, in gray. The behavior of the two models is comparable, but RANa achieves higher values in both metrics.

**Examples of retrieved context** – Fig. 6 shows three types of context for $N = 8$. *Top 8* most similar dataset images to the goal in DINOv2 feature space (top) and *top 8 MMR* most similar images filtered by MMR (middle), are *static* context types, *i.e.*they are fixed along one episode and depend only on the goal image, shown on the right highlighted in red. *DINOv2-Graph* (bottom) is a *dynamic* context, recalculated at each step and depending on the observation (left, highlighted in blue).

The top 8 most similar images to the goal (top) are very similar to each other as well, and might not be very useful. Filtering images using MMR increases diversity and performance (cf. Appendix D for a qualitative analysis). The *dynamic* context (bottom) is the most informative, as it provides some form of *"waypoints"*

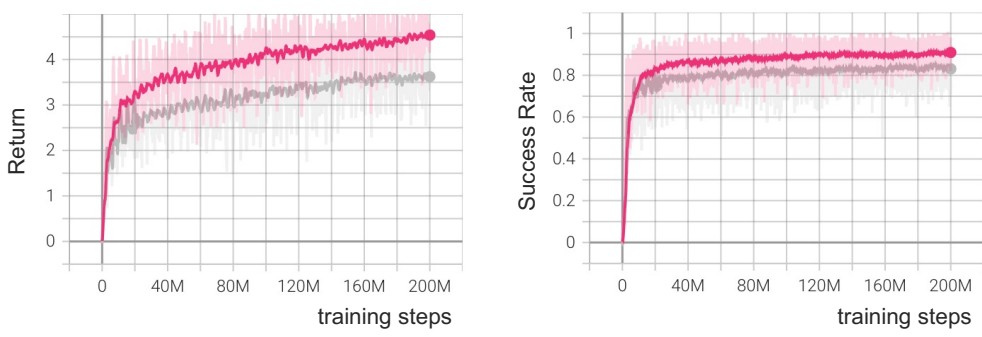

Figure 5: **Training curves**: Return and Success Rate for RANa-t *static* (pink) and DEBiT (gray).

*Static*, *top 8*

*Static*, *top 8 MMR*

*Dynamic*, *DINOv2-Graph*

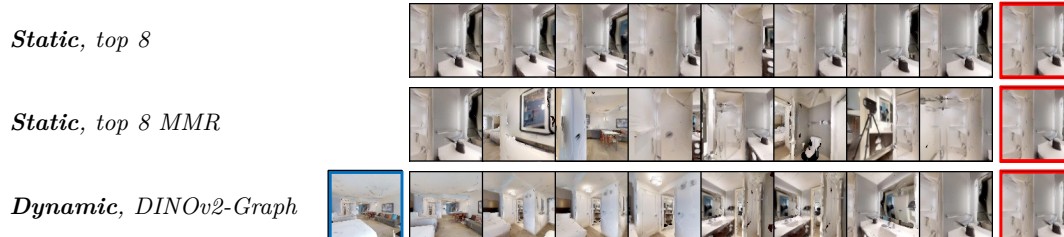

Figure 6: **Context types** - *top 8* most similar dataset images to the goal (red, on the right), *top 8 MMR*, most similar images filtered by MMR, and *dynamic*, the only one depending on the observation (blue, left) as well.

| Model | Extra sensors | ObjNav | | IIN | |
|---|---|---|---|---|---|
| | | SR | SPL | SR | SPL |
| ZSEL (Al-Halah et al., 2022) | – | 11.3 | – | – | – |
| ZSON A (Majumdar et al., 2022) | – | 31.3 | 12.0 | 16.1 | 8.4 |
| Mod-INN (Krantz et al., 2023)† | Depth, odom | – | – | 56.1 | 23.3 |
| **DEBiT-b** | – | 31.9 | 15.8 | 55.4 | 25.5 |
| **RANa-b** | – | **36.6** | **17.9** | **58.7** | **26.8** |

Table 8: **Zero-shot navigation results using retrieved image goal** $\mathbf{g}^{\mathcal{R}}$ for Gibson *ObjectNav* and HM3D *Instance-ImageNav* , and comparison to state-of-the-art. † obtained with camera intrinsics and robot size of the original *Instance-ImageNav* task specification.

from the current location to the goal – indeed this approach achieves the best navigation performance, as shown in Tables 2 and 3.

## 5.2 Retrieval allows zero-shot generalization

By retrieving the image goal $\mathbf{g}^{\mathcal{R}}$ from the database, it is possible to apply any *ImageNav* model to the *ObjectNav* and *Instance-ImageNav* tasks. Table 8 compares the performance of zero-shot architectures in: (i) *Gibson ObjectNav*, a variant introduced in Al-Halah et al. (2022) for *ImageNav* agents, consisting of 1, 000 episodes over 5 Gibson scenes and containing the 6 object categories of HM3D *ObjectNav* (*chair, bed, plant, sofa, tv, toilet*), and (ii) HM3D *Instance-ImageNav* (Krantz et al., 2023) with modified camera intrinsics and robot size to fit the *ImageNav* configuration. In both cases we use retrieval databases of 50, 000 images, as it is important to retrieve an image goal $\mathbf{g}^{\mathcal{R}}$ as close as possible to the real goal to avoid stopping too far from it. We ablate this parameter in Appendix D and show that RANa already achieves SOTA zero-shot *Instance-ImageNav* performance with as little as 5, 000 retrieval images. In *Instance-ImageNav* we replace the image goal with the closest image in DINOv2 feature space from the retrieval set. In zero-shot *ObjectNav*, where the goal is an object category, we retrieve the top 9 closest dataset images using OpenCLIP (Ilharco et al., 2021) feature similarity, and re-rank them *at each step* to select as goal $\mathbf{g}^{\mathcal{R}}$ the element with the highest co-visibility with the current observation (the remaining 8 images constitute the context). This is achieved using the frozen co-visibility auxiliary loss head that was used to train DEBiT models. This step is needed since in *ObjectNav* any instance of a target object is a valid goal.

Retrieval enables the use of DEBiT-b for both tasks, achieving strong performance in line with best existing methods (for reference, zero-shot DEBiT-b without retrieved goal image achieves 10 SR and 3.1 SPL on *Instance-ImageNav*). RANa outperforms the feature-matching based method of Krantz et al. (2023) which uses additional depth and odometry sensors and is the current SOTA in zero-shot *Instance-ImageNav*, as well as ZSON (configuration A) of Majumdar et al. (2022) in zero-shot *Gibson ObjectNav*.

Fig. 7 shows an example of successful zero-shot *Instance-ImageNavigation*, where our RANa agent first correctly picks from the context the door near the goal ①, and then a view of the room where the target is ②, leading to success. In contrast, the DEBiT-b baseline without context does not see the target inside the room and passes by, exploring the wrong side of the house and failing to complete the episode.

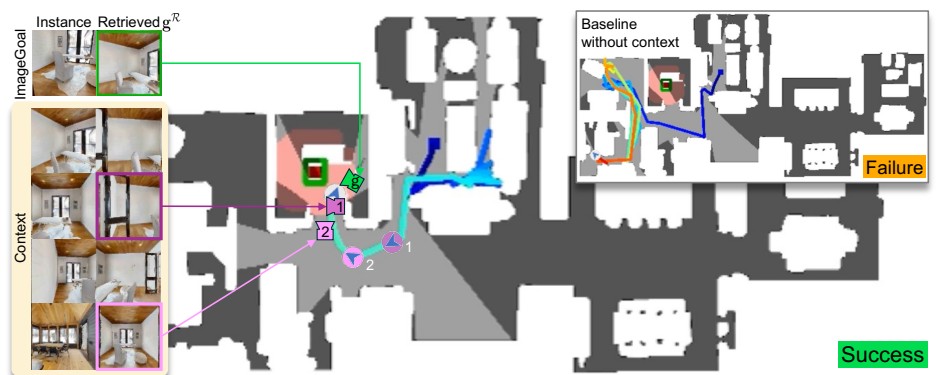

Figure 7: **Zero-shot *Instance-ImageNav* episode** by RANa-b and by DEBiT-b baseline (top-right box). The green camera indicates the pose of the retrieved image goal $\mathbf{g}^{\mathcal{R}}$, the pink ones indicate the poses of selected context items.

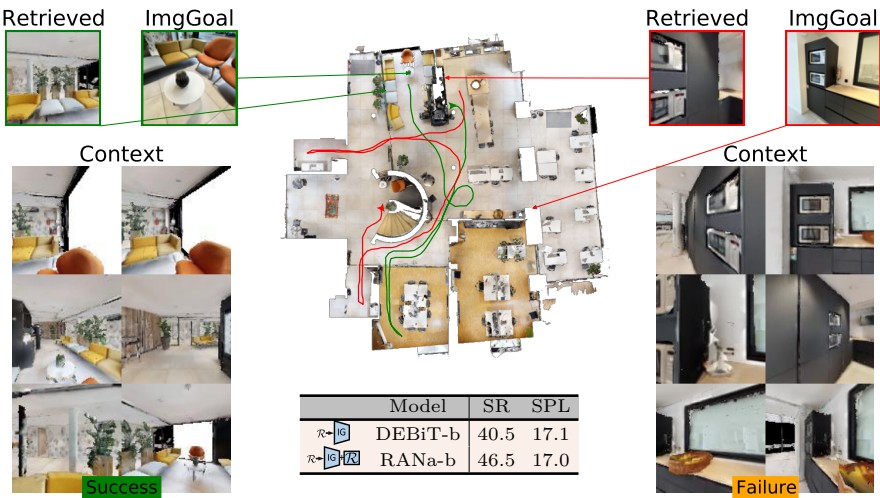

Figure 8: **Navigation with real goal images**, retrieving new goals: 2 episodes shown, 200 episodes evaluated.

## 5.3 Retrieval allows bridging the sim-to-real gap

We evaluate our ℛ⊷ⒾⒼ⊢ℛ RANa-b agent on a small *Instance-ImageNav* dataset of 200 episodes where goal images are taken in a *real* office environment. We manually annotate the positions of 20 real images in a simulated 3D reconstruction of the environment. We generate a retrieval set $\mathcal{D}$ of 10k images from uniformly-sampled navigable poses, rendered in simulation using the agent configuration. Simulating navigation from 10 random starting poses towards each of the 20 real image goals, RANa reaches 46.5 success, vs. 40.5 of DEBiT-b (see inlet Table in Fig. 8). The Figure illustrates a challenging success case (left), where the cactus viewing angle is far from any view the agent could capture, and (right) an interesting failure case, where the retrieval returned as goal $\mathbf{g}^{\mathcal{R}}$ an image of a different instance of the same microwave model at a different position. Interestingly, some retrieved context images depict the correct goal, and RANa appears to be able to exploit this information as it succeeds in 3/10 episodes featuring this goal.

## 6 Extension to *ObjectNav*

We extend the proposed approach to *ObjectNav*, where the goal is a textual label of an object category, by slightly modifying the RANa architecture. Since *ObjectNav* requires semantic understanding of the scene and common-sense reasoning, a large body of research builds semantic maps (Gadre et al., 2022; Chaplot et al., 2020a) and/or leverages an LLM to guide navigation based on a scene's common sense semantics (Yu et al., 2023; Cai et al., 2024; Kuang et al., 2024; Yin et al., 2024). Few approaches address this task in a

| Model | Extra sensors | w/ map | w/ LLM | SR | SPL |
|---|---|---|---|---|---|
| PixNav (Cai et al., 2024) | Pano RGB | ✗ | ✓ | 37.9 | 20.5 |
| ESC (Zhou et al., 2023) | Depth, odom | ✓ | ✓ | 39.2 | 22.3 |
| L3MVN (Yu et al., 2023) | Depth, odom | ✓ | ✓ | 54.2 | 25.5 |
| CoW (Gadre et al., 2022) | Depth, odom | ✓ | ✗ | 6.1 | 3.9 |
| GoW (Gadre et al., 2023) | Depth, odom | ✓ | ✗ | 32.0 | 18.1 |
| ProcTHOR (Deitke et al., 2022) | Depth, odom | ✓ | ✗ | 13.2 | 7.7 |
| OVRL-v2 (Yadav et al., 2023a) | Odom | ✗ | ✗ | 64.7 | 28.1 |
| ZSON (Majumdar et al., 2022) | – | ✗ | ✗ | 25.5 | 12.6 |
| PSL (Sun et al., 2024b) | – | ✗ | ✗ | 42.4 | 19.2 |
| **DEBiT-t** *ObjectNav* | – | ✗ | ✗ | 42.9 | 23.0 |
| **RANa-t** *ObjectNav* | – | ✗ | ✗ | **52.1** | **26.9** |

Table 9: **DEBiT-t and RANa-t *ObjectNav* compared to state-of-the-art methods.**

zero-shot fashion, training their agents without *ObjectNav* rewards and leveraging CLIP encoders for goal representation (Sun et al., 2024b; Majumdar et al., 2022; Gadre et al., 2023). Here we do not aim to achieve state-of-the-art results, as it is unlikely for our simple approach to compete with methods using LLMs and VLMs with billion of parameters. Instead we want to showcase the effectiveness and flexibility of our method.

In order to do that, we simply modify few components of the RANa-t *static* architecture: to account for the textual nature of the goal we use the multi-modal FM OpenCLIP (Ilharco et al., 2021) to compute the goal representation $\tilde{\mathbf{g}}_t$ and the similarity measure used for retrieval. Besides, *ObjectNav* has two additional actions, LOOK UP and LOOK DOWN, that are added to the action space. We train and test this agent, as well as a baseline not augmented with retrieval (we refer to it as DEBiT-t *ObjectNav*), on the HM3DSem dataset (Ramakrishnan et al., 2021; Yadav et al., 2023c) with 6 object categories (*chair, bed, plant, sofa, tv, toilet*) using the standard Habitat *ObjectNav* task definition and following the same procedure described earlier.

Table 9 compares the results of RANa-t and DEBiT-t *ObjectNav* models with competing approaches. RANa can exploit retrieved context also in this setting and improves the DEBiT baseline SR by almost +10 points. Compared to methods with similar sensor settings (only one RGB camera), RANa-t outperforms previous SOTA by +9.7 on SR and +7.7 on SPL. We also compare RANa-t to methods that use extra sensors, such as depth camera, GPS+compass, or 6 camera sensors, as well as LLM-guided navigation. Here RANa-t shows competitive results, outperforming several strong baselines such as ESC (Zhou et al., 2023) and PixNav (Cai et al., 2024).

## 7 Conclusion

We propose a general retrieval-augmented navigation agent, trained with RL, that is able to retrieve and act on images stored in a large, global database of robot observations. We leverage pre-trained foundation models to allow the agent to query and process contextual information that can be useful for solving its target task. Augmenting a SOTA *ImageNav* agent with a context retrieval module improves navigation performance while adding minimal overhead, which showcases the flexibility and effectiveness of the proposed method. Moreover, we demonstrate the approach for an *ObjectNav* agent, and we show how to use retrieval across domains and modalities to apply existing agents to new tasks in a zero-shot fashion.

This work opens the possibility of letting navigation agents benefit from the large amount of unstructured data collected during the operation of robot fleets. Future work will explore the integration of metadata to the database and data-driven query mechanisms that allow an agent to target the information it needs.

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

## A  Retrieval dataset creation

We use the Habitat simulator, platform and task definitions from Savva et al. (2019). For *ImageNav*, we use the Gibson dataset (Xia et al., 2018), consisting of 72 train and 14 eval scenes, and the *ImageNav* agent configuration: height 1.5m, radius 0.1m, 112×112 RGB camera with 90° horizontal field-of-view (hfov) at 1.25m from the ground. To train and test the *ObjectNav* variant in Section 6, we use 80 train and 20 eval scenes of the HM3DSem-v0.2 (Ramakrishnan et al., 2021; Yadav et al., 2023c) as in Cai et al. (2024) and an agent of height 0.88m, radius 0.18m with 112×112 RGB sensor with 79° hfov placed at 0.88m from the ground.

For each scene, we generate a retrieval dataset $\mathcal{D}$ usually containing $1,000$ FPV images as follows:

  (i) we randomly choose start and goal viewpoints such that the goal is navigable from the start,

  (ii) we make an agent follow the shortest path from start to goal, and

  (iii) after each action, we save the agent's RGB observation as an image in $\mathcal{D}$.

We repeat this process until the dataset contains the desired number of images. We randomly choose the distribution of starts and goals such that the agent roughly covers all navigable areas, and that the path between pairs of viewpoints are not too close to one another to simulate a prior offline exploration of the scene.

## B  RANa agent architecture

In this section we detail some of the architecture choices made for our method.

**Observation encoder** – $x$ processes the RGB input image $\mathbf{x}_t \in \mathbb{R}^{3 \times 112 \times 112}$. It is implemented with a half-width ResNet-18 (He et al., 2016) and generates an output feature $\tilde{\mathbf{x}}_t \in \mathbb{R}^{512}$.

**Goal comparison function** – $g$ compares the observation $\mathbf{x}_t$ to the goal $\mathbf{g}$. For *ImageNav* we use the frozen binocular encoder of DEBiT (Bono et al., 2024a), based on the Geometric Foundation Model (FM) CroCo (Weinzaepfel et al., 2022). This encoder is a Siamese encoder $E$ applied to both $\mathbf{x}_t$ and $\mathbf{g}$, and a decoder $D$ which combines the output of the two encoders. Both $E$ and $D$ are implemented with a ViT architecture with self-attention layers, and $D$ also adds cross-attention. The output of the decoder $D$ is further compressed by a fully connected layer (frozen, from the DEBiT model) that projects the flattened output of $D$ into $\tilde{\mathbf{g}}_t \in \mathbb{R}^{3136}$. For details about the binocular encoder $g$ we refer to Bono et al. (2024a) and Weinzaepfel et al. (2022).

In *ObjectNav* we use OpenCLIP (Ilharco et al., 2021) to encode the visual observation $\mathbf{x}_t$ and the textual goal $\mathbf{g}$. This process produces two features $cl_V(\mathbf{x}_t) \in \mathbb{R}^{512}$ and $cl_T(\mathbf{g}) \in \mathbb{R}^{512}$, which are concatenated to generate the output $\tilde{\mathbf{g}}_t \in \mathbb{R}^{1024}$.

**Action embedding** – $l$ encodes the previous action $\mathbf{a}_{t-1}$ into a 32-dimensional feature vector $\tilde{\mathbf{a}}_t$ and it is taken from the DEBiT (Bono et al., 2024a) model. For *ImageNav*, we load DEBiT action encoder and freeze it during training, while for *ObjectNav*, we trained it from scratch, since the two tasks do not share a common action space. Indeed, the task definition of *ObjectNav* requires two additional actions, LOOK UP and LOOK DOWN.

**Gumbel context encoder** – **Variant 1** of the trainable context encoder $c$ (Section 3.1) selects one element $\tilde{\mathbf{c}}_t$ among the context features $\{\mathbf{e}_{t,n}\}_{n=1}^{N}$ based on a learned *relevance* estimate $\{\alpha_n\}_{n=1}^{N}$. The retrieved context images are compared to the goal image (if available) and to the observation by the same frozen binocular encoder of DEBiT described above, leading to the $N$ context features $\{\mathbf{e}_{t,n}\}_{n=1}^{N}$ each of dimension $\mathbb{R}^D$, where $D = 6272$ for *ImageNav* or *Instance-ImageNav* (when goal image is available), and $D = 3136$ for *ObjectNav*. A fully connected layer Linear($D, 1$) (using a syntax inspired by PyTorch) estimates the relevance $\{\alpha_n\}_{n=1}^{N}$ for each context element and the item with the largest relevance is selected using the Gumbel soft-max sampler (Jang et al., 2017). In the *ImageNav* or *Instance-ImageNav* cases, only the binocular feature representing the comparison between observation and context items is selected, so in all cases the context feature $\tilde{\mathbf{c}}_t \in \mathbb{R}^{3136}$.

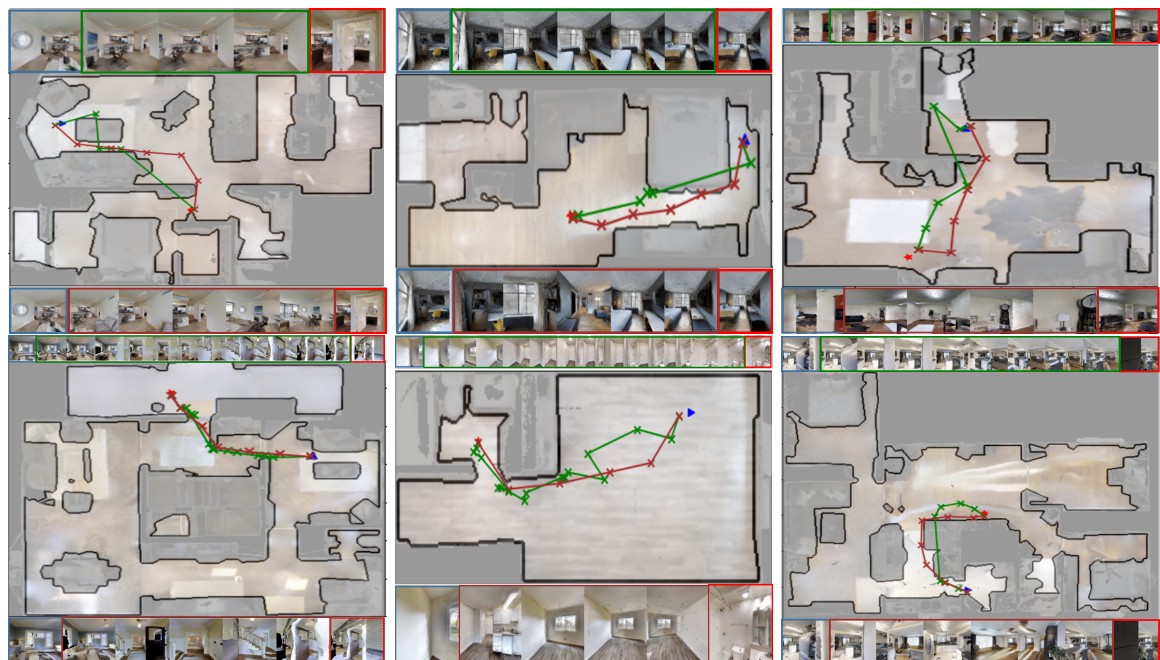

Figure 9: **Examples of shortest paths on DINOv2-Graph (SWG) and on Pose Graph (PS)** - The most similar image $r_1^{\mathbf{x}_t} \in \mathcal{D}$ to the goal is highlighted in red and localized with a red star on the floorplan, while $r_1^{\mathbf{g}} \in \mathcal{D}$, in blue, is the most similar image to the current observation. Green crosses connects images along the shortest-path in the DINOv2-Graph (corresponding images are highlighted in green on top of the map) and purple crosses refer to poses on the Pose Graph, with corresponding images shown with purple contour at the bottom of the map.

**Attention-based context encoder** – **Variant 2** of the trainable context encoder creates a context feature by computing the self-attention over the context elements $\{\mathbf{e}_{t,n}\}_{n=1}^N$ computed as described in the previous paragraph. In this case the context encoder $c$ first reduces the dimensionality of each context feature to $\mathbb{R}^{128}$ through a fully connected layer $\texttt{Linear}(D, 128)$ ($D = 6272$ for *ImageNav* or *Instance-ImageNav*, and $D = 3136$ for *ObjectNav*), then processes them with a standard two-layer transformer encoder network (Vaswani et al., 2017) with 4 attention heads and feedforward dimension 256. The $N$ context features are concatenated into a feature vector of size $\mathbb{R}^{N \times 128}$ and processed by a 2-layer $\texttt{MLP}(128, 3136)$ with hidden dimension 128 and output 3136, such that the context encoder output $\tilde{\mathbf{c}}_t$ has the same dimension as the Gumbel context encoder.

**Recurrent memory** – The flatten vectors $\tilde{\mathbf{x}}_t$, $\tilde{\mathbf{g}}_t$, $\tilde{\mathbf{a}}_t$ and $\tilde{\mathbf{c}}_t$ are concatenated and fed to a single-layer GRU (Cho et al., 2014) with hidden state $\mathbf{h}_t \in \mathbb{R}^{512}$. This GRU is an "extended" version of the baseline agent's GRU without context, which is finetuned. However, since RANa concatenates the context feature $\tilde{\mathbf{c}}_t$ to the other inputs, we cannot directly finetune the GRU as is. We modify the GRU of the base model by padding the input weight matrices of the first layer, $W_{ir}^0, W_{iz}^0, W_{in}^0$ (and corresponding bias vectors $b_{ir}^0, b_{iz}^0, b_{in}^0$) along the input dimension with random values. The remaining GRU matrices are initialized to the values of the baseline checkpoints and finetuned from there.

**Policy** – The hidden state $\mathbf{h}_t$ is fed to a linear policy $\pi$ composed of a linear *Actor* head that generates a softmax distribution over actions and a linear *Critic* head that evaluates the current state.

## C   DINOv2-Graph

In this section we provide details about the construction of the DINOv2-Graph context. The graph is built from the affinity matrix $\Omega_{\mathcal{D}}$ containing the similarity between DINOv2 (Oquab et al., 2024) features of all pairs of images in $\mathcal{D}$, where the nodes are images $\mathbf{x}_i^{\mathcal{P}} \in \mathcal{D}$ and edges are weighted by similarity. We use Dijkstra's algorithm (Dijkstra, 1956) (from the SciPy python package) to select the shortest path between two nodes.

|  | SWG | SBG | DWG | PG |
|---|---|---|---|---|
| RANa-b (Gumbel) | **89.7** (*60.4*) | 89.6 (*60.5*) | 88.7 (***60.7***) | 89.7 (*60.7*) |
| RANa-b (Attention) | 89.8 (*71.0*) | 89.6 (*70.3*) | **90.7** (***71.8***) | 91.9 (*69.8*) |
| RANa-t (Attention) | 84.7 (*63.7*) | 83.5 (*61.1*) | **86.8** (***64.8***) | 87.4 (*62.2*) |

Table 10: 🇬🇷 **Impact of the graph structure on RANa with DINOv2-Graph** - SR (*SPL*) for *ImageNav* with retrieval using different graph structures at `test` time: Sparse Weighted Graph (SWG), Sparse Binary Graph (SBG), Dense Weighted Graph (DWG) and Pose Graph (PG). All RANa models were trained with the same SWG graphs built on the training scenes. In **bolded best** results. Note that the PG variant is not considered as it uses the camera poses of images in $\mathcal{D}$; we only add it for comparison.

Selecting a path on this semantic graph (DINOv2-Graph) is not necessarily equivalent to a "physical path" (see examples in Fig. 9). In order to enforce selection of edges between images that have high overlap probability (*i.e.* represent the same part of the scene) we made two further modifications to the above graph. First, as we observed that a DINOv2 similarity score above $th = 0.75$ between L2 normalized features reflects, in most cases, a good overlap between the two images, we only kept edges for which the similarity was above this threshold, removing all the others. Second, in order to enforce Dijkstra's algorithm to select paths with high similarity (low cost), we used the weights $w_{ij} = \sqrt{(1 - s_{ij})}$, where $s_{ij}$ is the DINOv2 similarity between the L2 normalized features.

During both training and inference, at each step, the agent finds in $\mathcal{D}$ the most similar images $r_1^{\mathbf{x}_t}$ and $r_1^{\mathbf{g}}$ to the observation $\mathbf{x}_t$ and goal $\mathbf{g}$, respectively, and computes the shortest path between $r_1^{\mathbf{x}_t}$ and $r_1^{\mathbf{g}}$ on the DINOv2-Graph. Let $\mathcal{P} = \{r_1^{\mathbf{x}_t}, r_1, r_2, ..., r_n, r_1^{\mathbf{g}}\}$ be the set of nodes on the shortest path and $C$ the context size. If $n + 2 > C$, we sample images randomly from $\mathcal{P}$ and update all images in the context. If $n + 2 \leq C$, we use all images in $\mathcal{P}$ to update $n + 2$ images in the context. If no path was found, $n = 2$ hence we only update two elements of the context with $r_1^{\mathbf{x}_t}$ and $r_1^{\mathbf{g}}$.

**Analyzing the choice of the graph structure** – We trained our models on the Gibson dataset where we had around 10k images per scene in the retrieval dataset $\mathcal{D}$. We selected the checkpoint obtained at 130M steps for all models. During testing we used a set of $|\mathcal{D}| = 1$k images per scene. In this section we study how the graph structure affects performance. The default model, used for the results reported in Table 2 of the main paper, builds a weighted graph by removing edges with similarity below $th = 0.75$ and uses $w_{ij} = \sqrt{(1 - s_{ij})}$ as weights. We call this Sparse Weighted Graph (SWG). We evaluate three additional strategies to construct the DINOv2-Graph. (i) We consider the same graph, but instead of using $w_{ij} = \sqrt{(1 - s_{ij})}$ we use $w_{ij} = 1$ as edge weights, and we call this model Sparse Binary Graph (SBG). (ii) We also consider a similarity threshold $th = 0.4$ which yields a much denser graph, called Dense Weighted Graph (DWG). (iii) Finally, we compare these results with a "pose graph", where instead of using DINOv2 features, we use distances between camera poses to build the affinity map. In this case we consider a binary graph where we cut edges for which the distance between the camera poses is above 1m. We denote the Pose Graph with PG.

From the results shown in Table 10 we can draw the following observations. First, using a pose aware graph (PG) does not necessarily improve the accuracy over using DINOv2-Graph, and when it does, the improvement is rather small. Note however that the pose of the observation and goal images were not used to select the start and end points of the shortest path, $r_1^{\mathbf{x}_t}$ and $r_1^{\mathbf{g}}$, these were retrieved using DINOv2 feature similarity and we only replaced the DINOv2-Graph with the Pose Graph. The second observation we can make is that the graph structure does not have a big impact on the model with a Gumbel selector, but a bigger one on the model with attention-based context encoder. This is not surprising as the Gumbel selects an image from the context (path) and use it independently from other context elements, while Attention combines information from all images within the context (path), hence it is more important how we fill up the context. Finally, we observe that the Dense Weighted Graph (DWG) performs better than the Sparse Weighted Graph (SWG). We conjecture that the reason is that the model was trained with $\mathcal{D}$ of size 10k images and we rarely had no connection between two requested nodes ($r_1^{\mathbf{x}_t}$, $r_1^{\mathbf{g}}$), which occurs more often at test when only 1k images are available. Using a denser graph (DWG) appears to help overcoming these cases.

| context size | 4 | 8 | 12 | 16 | 20 | 24 |
|---|---|---|---|---|---|---|
| 4 | 80.8 (*52.3*) | 80.5 (*51.6*) | 80.4 (*52.7*) | 81.0 (*52.3*) | 80.9 (*52.0*) | 81.7 (***54.0***) |
| 8 | 80.4 (*49.8*) | 83.0 (*52.3*) | 80.6 (*50.6*) | **84.0** (*53.0*) | 82.0 (*53.2*) | 82.9 (*53.4*) |
| 12 | 81.9 (*50.0*) | 80.0 (*49.2*) | 81.9 (*51.2*) | 80.6 (*50.6*) | 81.6 (*50.8*) | 82.4 (*52.2*) |

Table 11: 🔲↔ℛ **Impact of context size for RANa-t** - SR (*SPL*) for *ImageNav* with retrieval over different context sizes, combinations of train and test conditions.

| context size | 4 | 8 | 12 | 16 | 20 | 24 |
|---|---|---|---|---|---|---|
| 4 | 51.6 (*26.1*) | 50.8 (*25.7*) | 49.8 (*25.3*) | 49.8 (*25.3*) | 49.0 (*24.7*) | 50.6 (*25.8*) |
| 8 | 50.5 (*25.9*) | 51.9 (*26.4*) | **52.1** (*26.9*) | 50.3 (*26.6*) | 51.6 (*26.5*) | 51.1 (*26.6*) |
| 12 | 51.4 (26.4) | 51.4 (26.9) | 50.9 (26.7) | 51.2 (27.3) | 50.9 (26.7) | 51.6 (**27.4**) |

Table 12: 🔲↔ℛ **Impact of context size for RANa-t** - SR (*SPL*) for *ObjectNav* with retrieval over different context sizes, combinations of train and test conditions.

**Similarities and differences with topological maps** – The proposed graph representation is related to topological map structures (Savinov et al., 2018; Beeching et al., 2020b; Chaplot et al., 2020c; Wiyatno et al., 2022; Kim et al., 2023; Sridhar et al., 2024). However, there are several critical differences between our model and traditional topological map based methods. First, we do not use the topological map to localize the agent, plan, and act to reach the "next waypoint". Instead, the information gathered from planning is passed to the agent as a "recommendation" in the form of the retrieval context, to be integrated into decision making through learning: our agent is a full fledged policy which has access to memory, goal and a series of images along the path, which arguably enables more informed decision-making, and improved robustness and navigation performance. Additionally, our graph is connected based only on visual similarity rather than predictions from temporal distances. While it is debatable whether this approach is inherently superior, it offers a distinct advantage: we do not require trajectories to build the graph but we can also build it from a large number of isolated images. The main drawback of our method is that we rely on the assumption that visual proximity is a proxy for spatial proximity, which is not always true (see Fig. 9). Future work could attempt to learn to take into account the uncertainty in this assumption through data-driven planning, as in Beeching et al. (2020b).

# D   Ablation experiments

In this section, we provide additional ablation studies that could not fit into the main document. As for the ablation experiments in the main paper, all experiments are conducted on RANa-t models.

**Impact of context size** – is studied in Table 11 for *ImageNav* (🔲↔ℛ) and in Table 12 for *ObjectNav* (🔲↔ℛ) with *static* context. One interesting advantage of the Gumbel selector is that it allows for context of different sizes during evaluation, independent of the context size used during training. With that said, this parameter has a relatively minor impact on performance, with larger training context sizes showing positive performance trends. The size $N = 8$ is a choice that works well both for *ImageNav* and *ObjectNav* and balances navigation performance with execution time. For *ObjectNav*, we observe a more pronounced performance increase for larger context sizes during evaluation, so we use $N = 8$ during training and $N = 12$ during evaluation.

**Impact of context diversity for Gumbel context encoder** – Table 13 evaluates the behavior of RANa-t *ImageNav* with *static* context, when using different strategies to create the context, (i) *top 8 MMR*: most similar database images filtered by MMR as in Section 4.1, (ii) *top 8*: most similar database images to the goal in DINOv2 feature space, and (iii) *top 1*: the closest element from the retrieval dataset.

Filtering the context with MMR is important, as the agent appears to rely on diverse context items; removing MMR degrades performance. *top 1* always uses the best element from the context, which is very similar to the goal image. In this case performance go back to baseline levels, confirming the importance of diversity in the retrieved elements.

**Impact of retrieval database size** – on zero-shot *Instance-ImageNav* (ℛ↔🔲↔ℛ) is shown on Table 14. Here the goal image $\mathbf{g}^{\mathcal{R}}$ is retrieved from the database, and thus retrieval accuracy is crucial for success and

|  | top 8 MMR | top 8 | top 1 |
|---|---|---|---|
| top 8 MMR | **83.0 (*52.3*)** | 81.5 (*51.0*) | 79.6 (*48.7*) |
| top 8 | 80.7 (*51.1*) | 81.4 (*48.1*) | 80.2 (*51.2*) |

Table 13: **Impact of MMR filtering on retrieved context** - SR (*SPL*) for *ImageNav* with and without MMR, combinations of `train` and `test` conditions ( in-domain performance shaded in pink ). Context diversity from MMR is important and delivers the best performance (SR=83.0). When the model trained with MMR is tested without it, performance drops.

| DB size | 1k | 5k | 10k | 20k | 50k | 100k |
|---|---|---|---|---|---|---|
| SR (*SPL*) | 47.6 (*20.2*) | 56.2 (*24.4*) | 56.4 (*23.8*) | 57.4 (*25.0*) | 58.7 (***26.8***) | **59.1** (*25.3*) |

Table 14: ℛ⊸ᴵᴳ⊣ℛ **Impact of retrieval database size in zero-shot** - SR (*SPL*) in *Instance-ImageNavigation* increases with dataset size.

avoid terminating an episode too far from the actual goal. Table 14 shows this phenomenon, with zero-shot *Instance-ImageNav* performance increasing with retrieval dataset size. However, please note that RANa achieves navigation performance in line with SOTA using a retrieval dataset containing only $5,000$ images.

**Analysis of "oracle" context** – for *ImageNav* (ᴵᴳ⊣ℛ) is carried out in Table 15. In this experiment we study which information is most useful and can be exploited by RANa, by building contexts using privileged information in simulation – therefore in these cases the retrieval dataset $\mathcal{D}$ is not used. *Oracle 1* captures a panoramic view of the goal location and is created by collecting $N = 8$ context images at the goal position while rotating the camera view by 45° clock-wise for each element. This context represents a soft upper bound for the static context in Section 4.1, where a fixed set of items is chosen per episode based on their similarity with the goal image. *Oracle 2* context is richer and it is generated by computing the shortest path from the agent position to the goal at each step and collecting views along this path. This is a soft upper bound for the DINOv2-Graph dynamic context (Section 4.2), which is dynamically generated at each step on the context graph. Table 15 shows that, as expected, both types of oracle context can provide large gains, especially in SPL, *i.e.* navigation efficiency. It suggests that there is little room for improvement in success (SR) using static goal context, while more structured information, *e.g.* paths, derived from data in $\mathcal{D}$ can improve navigation considerably.

**Impact of using pre-trained GRU weights** – As mentioned earlier all RANa agents use a GRU finetuned from the baseline agent's GRU. We evaluate the impact of this choice by training the RANa-t *static* GRU from scratch while keeping all other settings fixed. Table 16 shows that finetuning the GRU form pretrained weights – those of DEBiT (Bono et al., 2024a) in this case – improves navigation performance. We also observe faster convergence during training, which might in part explain the performance increase.

**Context selection criteria** – All *ImageNav* experiments reported in the paper use a feature vector for each context element which is obtained by concatenating two features: (i) the output of the Geometric FM *g* comparing context element and observation, and (ii) the output of *g* comparing context element and goal. We evaluate the impact of this choice by training a RANa-t *static* model which selects context elements based only on feature (i), which compares context elements to the observation. The performance of this variant and that of the baseline which uses both observation *and* goal, is displayed in Table 17. It shows that information

| Context | Type | SR | SPL |
|---|---|---|---|
| DINOv2 goal | static | 83.0 | 52.3 |
| Panorama (*oracle 1*) | static | 83.4 | 71.6 |
| DINOv2-graph | dynamic | 84.7 | 63.7 |
| Shortest path (*oracle 2*) | dynamic | 98.3 | 87.9 |

Table 15: ᴵᴳ⊣ℛ **Navigation with "oracle" context** - SR (SPL) for RANa-t using privileged information to build oracle context.

| Model | SR | SPL |
|---|---|---|
| GRU finetuned | **83.0** | **52.3** |
| GRU from scratch | 81.4 | 51.0 |

Table 16: 🄸🄶⟶ℛ **Impact of GRU finetuning** - SR (SPL) for RANa-t with *static* context and GRU finetuned or trained from scratch.

| Model | SR | SPL |
|---|---|---|
| context selection based on obs *and* goal | **83.0** | **52.3** |
| context selection based on obs only | 81.0 | 50.3 |

Table 17: 🄸🄶⟶ℛ **Impact of context selection** - SR (SPL) for RANa-t with *static* context and elements selected based on comparisons with only the observation, or both goal and observation.

provided by comparing the context to the goal is useful and can be exploited by RANa to navigate more successfully and efficiently to the goal.

**Context building for *ObjectNav*** – Table 18 shows the behavior of RANa-t *ObjectNav* (🄾🄶⟶ℛ) when adopting different context building approaches, (i) *rand*: 8 random elements from the retrieval dataset, (ii) *top 8 $cl_S$*: most similar database images to the object goal encoder in CLIP feature space using CLIP similarity as metric, and (iii) *top 8 $\omega_\mathbf{o}$*: most similar database images using CLIP *score softmax normalization*. While CLIP allows to easily compare any open-vocabulary object category with retrieval gallery images, it has been observed in the context of zero-shot image classification that directly ranking images by similarity to the category is suboptimal (Qian et al., 2023). To address this issue for *ObjectNav*, we leverage the fact that we often target a pre-defined set of classes $\mathcal{O}$ (*e.g.* 6 in HM3D), and rescale the score of each image in $\mathcal{D}$ via softmax:

$$\omega_\mathbf{o}(\mathbf{x}) = e^{cl_S(\mathbf{o},\mathbf{x})} / \left( \sum_{\mathbf{o}' \in \mathcal{O}} e^{cl_S(\mathbf{o}',\mathbf{x})} \right)$$

where $cl_S(\mathbf{o}, \mathbf{x})$ denotes the CLIP similarity of image $\mathbf{x}$ and object category $\mathbf{o}$ and $\omega_\mathbf{o}(\mathbf{x})$ is its normalized similarity.

The observed behavior is similar to *ImageNav*. First, a RANa-t model trained with a context with randomly selected images from the scene gallery performs worse than a model trained with either top 8 images. Nevertheless, this model still outperforms our baseline trained without retrieval (SR=42.9 and SPL=23.0 for the non-retrieval baseline) and is more robust to random context images during evaluation. For RANa-t models trained with retrieval top 8 context, we observe improved performance when using softmax normalized scores.

**Impact of retrieval dataset size for *ObjectNav*** – is shown in Table 19. Results demonstrate that unlike *ImageNav* (Table 5), *ObjectNav* seems to need larger retrieval datasets both at training and at testing. Performance degrades significantly when the dataset $\mathcal{D}$ at training only has 1,000 images. At test, using larger retrieval datasets leads to better performance, with metrics degrading significantly with $\mathcal{D}$ containing 100 items.

**Impact of CLIP variants for *ObjectNav*** – is evaluated in Table 20. In this experiment, we build retrieval datasets $\mathcal{D}$ of size 10k images for each of the 100 HM3DSem-v0.2 scenes used for *ObjectNav* (80 train, 20 test). For each of the 6 target object categories (*chair, bed, plant, toilet, tv monitor, sofa*) and for each

| | rand | top 8 $cl_S$ | top 8 $\omega_\mathbf{o}$ |
|---|---|---|---|
| rand | 46.8 (*23.8*) | 47.7 (*25.0*) | 46.1 (*23.1*) |
| top 8 $cl_S$ | 44.3 (*23.0*) | 48.0 (*25.4*) | 48.5 (*26.0*) |
| top 8 $\omega_\mathbf{o}$ | 44.4 (*21.4*) | 51.3 (*25.9*) | **51.9 (*26.4*)** |

Table 18: 🄾🄶⟶ℛ **Impact of retrieval data selection** - SR (*SPL*) for RANa-t *ObjectNav* with different context building strategies, combinations of `train` and `test` conditions.

| DB size | 100 | 1k | 10k | 50k |
|---|---|---|---|---|
| 1k | 43.8 (21.7) | 45.3 (22.4) | 46.8 (23.2) | 47.1 (23.2) |
| 10k | 46.6 (*24.9*) | 50.3 (*26.4*) | **52.1** (*26.9*) | 51.8 (**27.2**) |

Table 19: **Impact of retrieval database size** - SR (*SPL*) for RANa-t *ObjectNav* with retrieval using different database sizes, combinations of train and test conditions.

| Vision-language model | softmax | top 1 | top 8 | top 20 |
|---|---|---|---|---|
| **OpenCLIP** (Ilharco et al., 2021) | ✗ | 40.6 | 76.0 | 88.5 |
| **OpenCLIP** (Ilharco et al., 2021) | ✓ | **78.1** | **95.8** | **97.9** |
| **CLIP** (Radford et al., 2021) | ✓ | 67.7 | 88.5 | 90.6 |
| **SigLIP** (Zhai et al., 2023) | ✓ | 70.8 | 91.7 | 94.8 |

Table 20: **Retrieval performance of CLIP variants**. Retrieval is considered success if the retrieved image is of the correct category *and* if it is closer than 1m to the target object.

scene, we do a search in the retrieval dataset $\mathcal{D}$ by querying the CLIP-encoded goal category (*i.e.* the CLIP text encoder for the sentence "a picture of a *category* inside a house") and comparing it with CLIP encoded dataset images of the scene, providing top 1, top 8 and top 20 success percentages. A retrieval is considered successful if the retrieved image (whose ground-truth pose is known), is closer to one of the viewpoints of a object instance of the correct category than the success threshold of the *ObjectNav* task, *i.e.* 1m. The Table shows that OpenCLIP performs better than alternative CLIP models, namely OpenAI's original CLIP weights (Radford et al., 2021) and SigLIP (Zhai et al., 2023), all with the softmax normalization being used. As observed earlier, ranking images by CLIP similarity to the category does not perform well (Qian et al., 2023); since the number of target classes is known (6 in all our experiments), rescaling the score of each image in $\mathcal{D}$ via softmax significantly boosts performance.

## E Qualitative analysis

### E.1 Examples of retrieved context for *ObjectNav*

Fig. 10 displays examples of context for RANa *ObjectNav* () using similarity in CLIP feature space for the six different goal categories, *chair, bed, plant, sofa, tv monitor* and *toilet*. Results accurately represent instances of the target object present in the scene, with the exception of the *tv monitor* class where we observe some errors, especially in scenes where monitors are absent.

### E.2 Navigation examples

In Fig. 11 we show examples of our agent RANa-b *static* navigating to two challenging *ImageNav* goals. The figures visualize the observation (left), goal (highlighted in green), $N = 8$ context elements, with the one selected at the given time step highlighted in red, and the agent path drawn on the scene floorplan on the right. Each row of pictures focuses on one time step along the episode and are ordered in time from top to bottom. In this figure and the following ones, time is not sampled uniformly: we selected time steps, typically towards the end of the episode, where the agent makes interesting context selections to solve the episode.

The episode in Fig. 11 (left) is challenging because the environment is rather complex, the goal image is ambiguous and it is "hidden" in a terrace at the extreme perifery of the navigable space. The baseline DEBiT-b agent, without context, does its best to explore the apartment and gets close to the terrace door where the goal is located, but does not manage to navigate to it. RANa-b selects from the context an image (highlighted in red) that contains the railing of the terrace that helps our agent solve this complex navigation episode. The navigation episode on the right is complex for similar reasons: large complex environment and goal image with very little information. RANa is capable of exploiting the few relevant context items, by first selecting the context image containing the door to the room where the goal is located, and then the cabinet, whose corner is visible on the goal image.

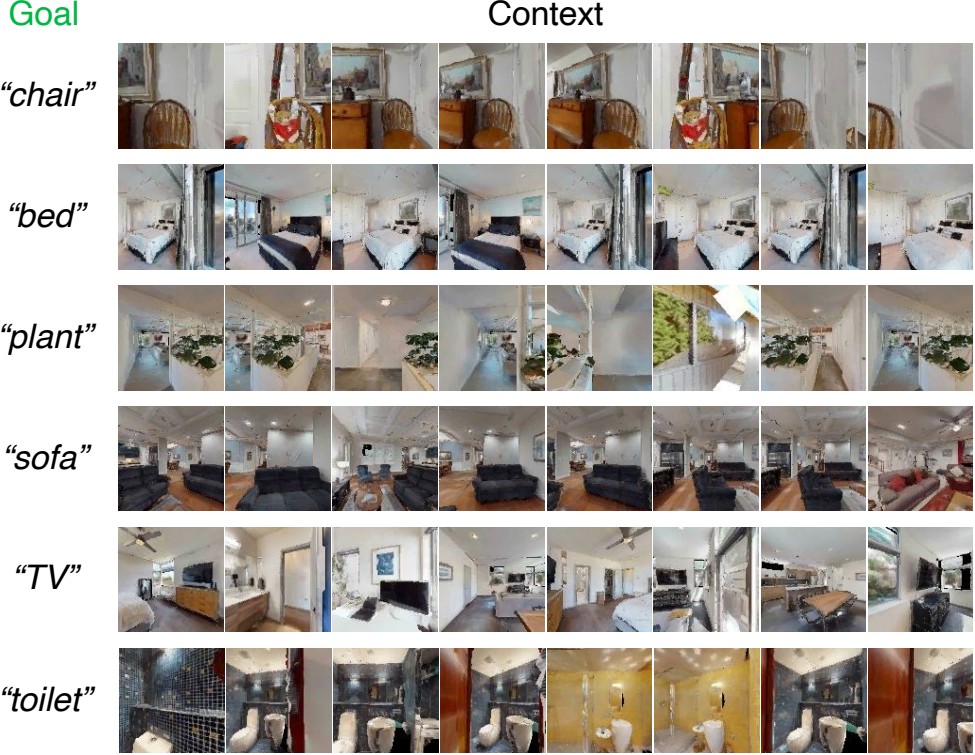

Figure 10: **Examples of context for different *ObjectNav* categories.**

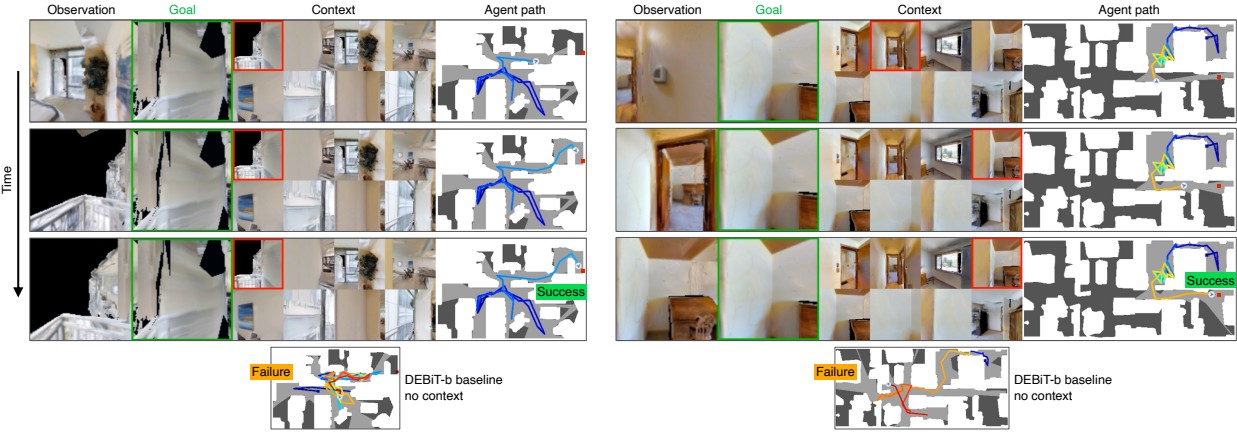

Figure 11: **RANa-b successfully navigates to challenging goals in two *ImageNav* episodes** - The agent observation is on the left, next to it is the goal image highlighted in green, then there are $N = 8$ context elements, with the one selected by the RANa agent at the current time step highlighted in red, and on the right the agent path drawn on the floorplan of the scene, with the goal position indicated by a red square. At the bottom we show the trajectory of the baseline DEBiT-b agent on the same episode.

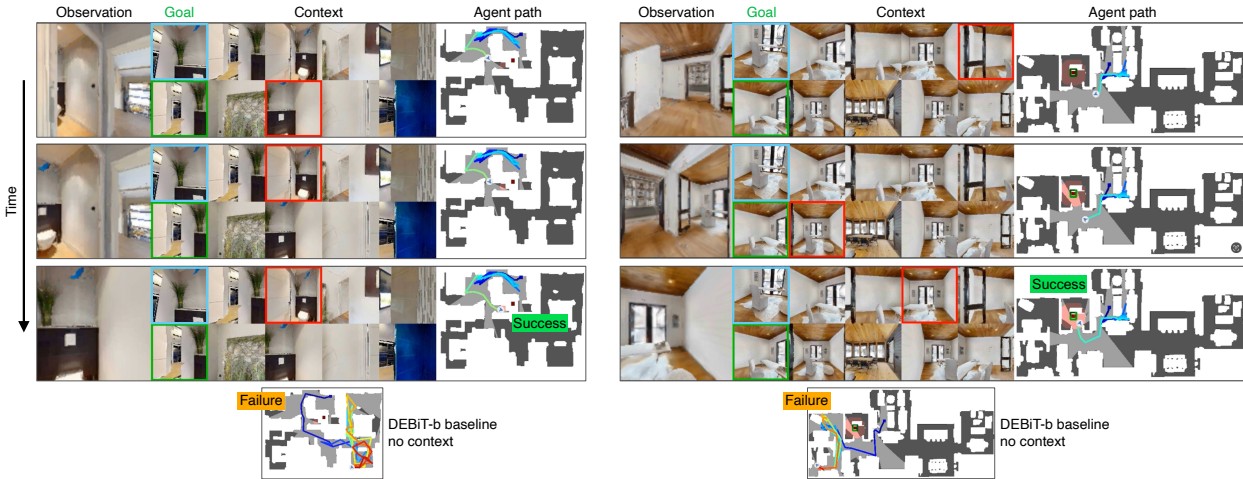

Figure 12: ℛ▸IG▸ℛ **RANa-b successfully navigating to two goals in zero-shot *Instance-ImageNav* episodes** - The agent observation is on the left, next to it is the original goal image from the *Instance-ImageNav* task (in cyan) and the retrieved goal image $\mathbf{g}^{\mathcal{R}}$ highlighted in green. The remaining data is organized and visualized as in Fig. 11.

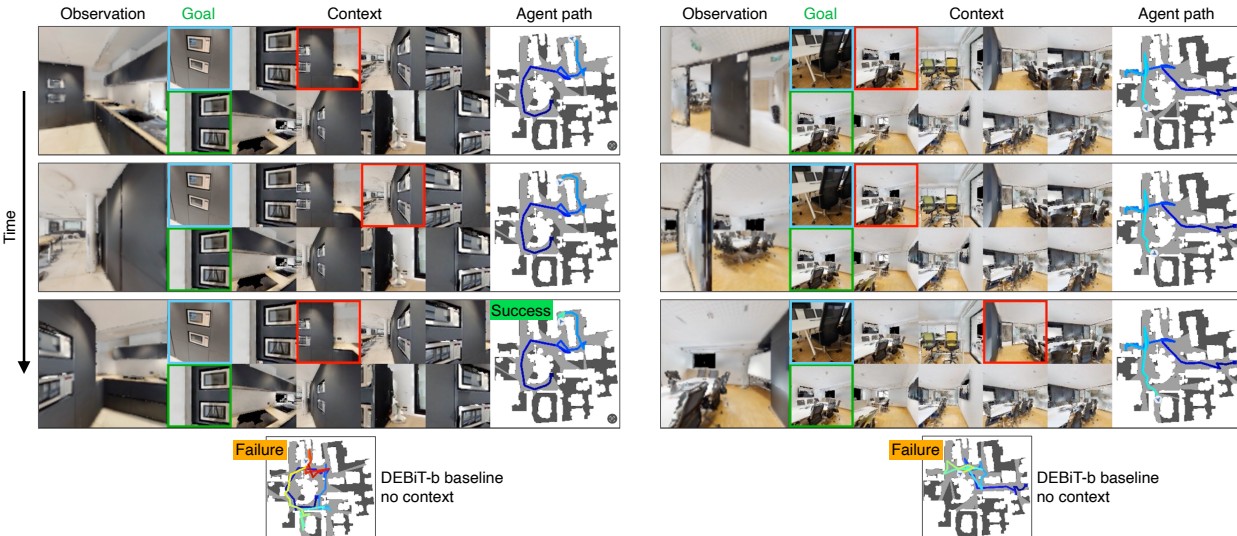

Figure 13: ℛ▸IG▸ℛ **RANa-b successfully navigates to real-world goals in two zero-shot *Instance-ImageNav* episodes** - In this case the original image goal (highlighted in cyan) is taken with *a smartphone in a real environment*. Again we retrieve a goal image $\mathbf{g}^{\mathcal{R}}$ from $\mathcal{D}$ and the agent navigates in a 3D model of the environment.

Fig. 12 shows RANa-b *ImageNav* agent successfully navigating in two zero-shot *Instance-ImageNav* episodes. Results are visualized in the same way as in Fig. 11, with the exception of the goal image. In this case we have in fact two goal images, one from the original task (highlighted in cyan) which is taken with a sensor with different pose and intrinsics of the agent sensor, and the goal image $\mathbf{g}^{\mathcal{R}}$, highlighted in green, which is retrieved from the scene database $\mathcal{R}$. In both cases, RANa selects context items that represent elements visible in our agent observation (the toilet on the left and the door on the right) that guide the agent inside the room where the goal is located. Interestingly, in both cases the baseline agent passed along the same path, but did not enter the room, arguably because it did not have access to the context.

In Fig. 13 we present an even more challenging scenario, where RANa navigates in two zero-shot *Instance-ImageNav* episodes, where the goal image is captured with *a smartphone in a real environment*. As mentioned in the main text, we assume having access to this environment and that a 3D model of the scene is available.

The agent navigates to the retrieved goal image $\mathbf{g}^{\mathcal{R}}$ from the database $\mathcal{D}$ in the simulated reconstruction of the scene. The example on the left is particularly interesting because the retrieved goal image, in green, is not the correct instance of the original goal. However, guided by several context items that contain instances of the original goal image, RANa succeeds in reaching the goal. In the navigation episode on the right, our agent picks relevant context elements containing the chairs of the meeting room where the goal is positioned, and enters the room solving the navigation episode.

