# OpenReview forum: "RANa: Retrieval-Augmented Navigation"
_TMLR — Accepted by TMLR_

### Review · Reviewer_mTWV · 2025-08-17

**Summary Of Contributions:**

**Summary of Contributions:**

This paper introduces a retrieval-augmented navigation (RANa) agent which can perform navigation tasks better by querying a database of previous observations and using it as additional contextual information.

The authors turn the simple idea of using a retrieval database into something powerful by storing observations as simple raw first person view (FPV) images and then using foundational models to encode that data.

The authors then introduce a set of benchmarks that provides a database of observations to evaluate retrieval-augmented agents. These benchmarks are based off standard Habitat Simulator tasks.

The authors also show that the RANa agent performs significantly better than other baseline models and also has zero-shot generalization capabilities.

**Strengths:**

- The paper is well-written and easy to follow
- The paper takes the simple idea of having a retrieval database and makes it into something powerful by making use of modern foundational models
- The authors also show that the database can be shared across agents, and is scalable to very large number of images
- Simplicity of the method: The authors store simple, raw images into the database without any complicated metadata or pose information and show that it can still work effectively by leveraging foundational models and a smart retrieval mechanism. This aligns well with the modern deep learning paradigm where large-scale raw data can be converted into useful representations.

**Weaknesses:**

- In the case of RANa agent, the performance of the agent will be closely tied to information obtained through the database. So it is very likely that there could be failure cases which cause the agent to go astray. The authors do briefly discuss about a failure case in Fig 12 in the Appendix but I feel that there should be more discussions on this.
- The retrieval basically works on visual similarity so one possible failure case could be if there are images in the database which are visually very similar but not relevant to the goal task. Wouldn’t the agent be led astray by the “additional context” in this scenario?
- One other interesting experiment to see would be the impact of “stale memory” in the agent’s database. For example, once a large enough samples for the database are collected and then the positions of the items in the simulator are modified slightly will the agent be led astray by the stale images in the database or is the agent robust enough to find the still find the goal. The authors do touch upon this a bit in the “microwave experiment” under the sim-to-real section but similar experiments could be directly performed in the simulator just to specifically test the robustness of the agent.

**Audience:**

Yes

**Audience Explanation:**

This paper introduces a unique method of foundational model based retrieval mechanism to significantly improve the performance of RL agents for the navigation task, so in my opinion it has novelty, strong results, and thus many people in TMLR's audience should be interested in this.

**Claims And Evidence:**

Yes

**Claims Explanation:**

The authors clearly support their claims but conducting extensive empirical experiments and presenting their results in the paper in section 5.

**Requested Changes:**

I feel that the paper is already strong enough to be accepted at TMLR, but it would be great if the authors could add a bit more discussion related to failure cases and robustness as mentioned in the weaknesses section.

---

> ### Author Response · Authors · 2025-09-05
> **Response to the Review**
>
> We are grateful to the Reviewer for the detailed and constructive feedback. We are happy to see that the Reviewer appreciated a well-written and easy to follow paper that presents a simple yet powerful and scalable method, that aligns well with the modern deep learning paradigm where large-scale raw data can be converted into useful representations. In the following we address all the Weaknesses and Requested Changes.
>
> > In the case of RANa agent, the performance of the agent will be closely tied to information obtained through the database. So it is very likely that there could be failure cases which cause the agent to go astray. The authors do briefly discuss about a failure case in Fig 12 in the Appendix but I feel that there should be more discussions on this.
>
> It is important to stress that the information obtained from the database is used to create a context, that can be viewed as a set of *suggestions* to help the baseline agent. This is in contrast with existing methods using e.g. topological maps, where database items are used as *waypoints*, which need to be accurate. Instead, the RANa agent learns to incorporate useful information in the policy, and discard useless one. This is supported by the experiments reported in Table 7, where we show that RANa performance does not degrade below baseline levels when providing random context images. In fact, we observe that only in few cases a potentially misleading context degrades performance: in other words, most failed episodes of RANa correspond to those of the baseline agent.
>
> > The retrieval basically works on visual similarity so one possible failure case could be if there are images in the database which are visually very similar but not relevant to the goal task. Wouldn’t the agent be led astray by the “additional context” in this scenario?
>
> While it is true that very similar but irrelevant context images might be more challenging than random ones, we observe that the model is in general robust to irrelevant context items, as shown in Table 7 where performance does not degrade below baseline levels when providing random context.
>
> > One other interesting experiment to see would be the impact of “stale memory” in the agent’s database. For example, once a large enough samples for the database are collected and then the positions of the items in the simulator are modified slightly will the agent be led astray by the stale images in the database or is the agent robust enough to find the still find the goal. The authors do touch upon this a bit in the “microwave experiment” under the sim-to-real section but similar experiments could be directly performed in the simulator just to specifically test the robustness of the agent.
>
> This is indeed an interesting and useful experiment to probe the model capabilities. However, we propose to defer it to future work where we extend the current set-up to interactive scenes. In fact the Habitat benchmark and scenes used in this work are static, i.e. furniture cannot be moved.
>
> As noted by the Reviewer, we have carried out sim-to-real experiments where dataset images do not exactly match the environment of the captured goal images. In this case we have not observed any strong effect due to the gap, arguably because the RANa agent learns to use or discard the database images in the context, depending on their usefulness.
>
> > I feel that the paper is already strong enough to be accepted at TMLR, but it would be great if the authors could add a bit more discussion related to failure cases and robustness as mentioned in the weaknesses section.
>
> Thanks for the encouraging comment and the recommendations.
>
> We have re-structured the paragraph entitled *Context is important* in Section 5, and added the discussion points raised above, including the failure case presented in the Appendix. The new paragraph is now titled *Context robustness and failure cases*.

---

### Review · Reviewer_ufQa · 2025-08-26

**Summary Of Contributions:**

The majority of the existing works consider robots to be new to the environment, without taking the prior observations and interactions into account. This work proposes retrieval-augmented navigation (RANa) to allow an agent to store and leverage visual observations from a retrieval database during operation. Retrieval-augmented agent involves a retrieval process which will retrieve the relevant context and the agent will make the next move based on the given context. The experimental results on ImageNav, Instance-ImageNav and ObjectNav demonstrate that RANa reaches new state-of-the-art results with a relatively fast inference speed.

**Audience:**

Yes

**Audience Explanation:**

See above.

**Claims And Evidence:**

Yes

**Claims Explanation:**

**Strengths**
1. The use of prior visual observations is smart and interesting.
2. The retrieval database, though small (i.e., 1K), brings significant performance improvements on ImageNav, Instance-ImageNav and ObjectNav.
3. The model can reach impressive zero-shot navigation performance.


**Weaknesses**
1. The success of RANa is highly dependent on the retrieval database. Can the model automatically update or add new visual observations into the retrieval database?
2. The database is still very small, and has limited scalability (Table 5). I understand that this may be attributed to limited context variety in the same environment. However, enlarging the retrieval database may be helpful for generalizability. Can the authors show the impact of retrieval database size on other datasets? If it does not bring improvement, then my question is: why doesn't a larger retrieval database bring further improvement?
3. What is the impact on inference speed with different sizes of retrieval database. Based on Table 5, the inference speed is significantly slower due to the DINOv2 computation (dynamic variant), and the affinity matrix is a significant computation cost (especially if a larger retrieval database is needed). I am also confused why DINOv2 is a computation-heavy component, given the elaboration on the last sentence in “Retrieval” subsection on Page 7.

**Requested Changes:**

1. More clarification is needed for “store” in Table 1

---

> ### Author Response · Authors · 2025-09-05
> **Response to the Review (1/2)**
>
> We would like to thank the Reviewer for taking the time to carefully review our paper and provide valuable feedback. We are glad that the proposed approach is found smart and interesting, bringing significant performance improvements on ImageNav, Instance-ImageNav and ObjectNav, and with impressive zero-shot navigation performance. In the following we address in detail the Weaknesses and Requested Changes.
>
> > The success of RANa is highly dependent on the retrieval database. Can the model automatically update or add new visual observations into the retrieval database?
>
> With the current formulation in which the retrieval database is not curated, we observe that we only need a reasonable spatial coverage of the scene. In the early stage of development we experimented with databases constituted with images collected along trajectories or with random views, and we did not notice any significant difference between the two strategies, as long as we had a reasonably uniform coverage of the environment.
>
> Our results indicate that the model is generally robust to various sizes of retrieval database, as well as the presence of irrelevant images, which suggests that adding new images as we explore the environment could be beneficial. Since we only need a single DINOv2 forward pass on the raw FPV image to add a new observation to the database, this would also be computationally feasible. It remains to be seen whether such a database construction process can reliably work from scratch, and whether some setups require curating the database as we go, e.g., by automatically detecting which images should be added or removed, especially if the environment changes over time. We plan to investigate this direction in future work.
>
> > The database is still very small, and has limited scalability (Table 5). I understand that this may be attributed to limited context variety in the same environment. However, enlarging the retrieval database may be helpful for generalizability. Can the authors show the impact of retrieval database size on other datasets? If it does not bring improvement, then my question is: why doesn't a larger retrieval database bring further improvement?
>
> Indeed, it might be possible that performance could increase by adding images to the database that do not belong to the scene but might be potentially relevant, e.g. to learn richer semantic representations or typical structures present in indoor environments. In this case, the gains of the RANa model would have a different nature and origin: instead of providing additional information on the scene, they would augment the general reasoning capacity of the agent.
>
> This could in principle be done, but in this case we would need to train a new model which would need to get this additional information during training.
>
> > What is the impact on inference speed with different sizes of retrieval database. Based on Table 5, the inference speed is significantly slower due to the DINOv2 computation (dynamic variant), and the affinity matrix is a significant computation cost (especially if a larger retrieval database is needed). I am also confused why DINOv2 is a computation-heavy component, given the elaboration on the last sentence in “Retrieval” subsection on Page 7.
>
> In the regime we are experimenting (100-100k images per scene), the impact on the inference speed of the dataset size is negligible - retrieval is extremely fast. For larger datasets, e.g. those covering an entire building, one might resort to dedicated indexing and retrieval methods like Faiss, that scale better with the dataset size.
>
> The reason for the higher inference time due to DINOv2 computation in the dynamic context case is that DINOv2 features need to be computed online for each observation to find the closest image in the dataset and estimate the shortest path on the affinity matrix. For images stored in the dataset, as customary, we pre-compute DINOv2 features, thus we do not include this process in the inference time. We have clarified this point in the caption of Table 1.
>
> Concerning the affinity matrix, it can indeed be expensive to compute, but it is only pre-computed once offline for the dataset, it is not re-computed during inference, only shortest paths are.
>
> Concerning the last sentence in “Retrieval” subsection on Page 7, what we meant is that we speed up  training with a trick to avoid computing DINOv2 features online. During *training only* we use privileged information in simulation about the agent pose to select from the database the nearest image, with its precomputed DINOv2 feature, and use it instead of the actual observation. This approximation allows to speed-up training considerably, but it is not used for inference as in principle we do not have access to this information.

---

> > ### Author Response · Authors · 2025-09-05
> > **Response to the Review (2/2)**
> >
> > > More clarification is needed for "store" in Table 1.
> >
> > Thanks for pointing this out.
> >
> > With "store" we mean that RANa only needs to save new images in a folder or database, without any additional information about pose, location etc., and without having to integrate the new observation in an existing structure like a map or a graph, which might be computationally expensive and prone to errors. For *static context* there is no other operation needed, although for performance reasons it might be preferable to pre-compute DINOv2 features (this is what we do in our experiments). In the case of *dynamic context* we also maintain a similarity matrix, which is computed offline. When new elements are added to the retrieval dataset, one row and one column for each new element is added to the matrix, representing similarities with all the other dataset items. We have added these clarifications in the description of Table 1.

---

> > ### Comment · Reviewer_ufQa · 2025-09-16
> >
> > Thanks for the rebuttal. It resolves majority of my concerns. For Response 2, I think it would be interesting to collect a "general" retrieval database, and see if it helps to generalize to novel environments.

---

> > > ### Author Response · Authors · 2025-09-17
> > > **Response to comment**
> > >
> > > > Thanks for the rebuttal. It resolves majority of my concerns.
> > >
> > > Thank you very much for staying engaged and for acknowledging the rebuttal, we are happy that it was helpful!
> > >
> > > > For Response 2, I think it would be interesting to collect a "general" retrieval database, and see if it helps to generalize to novel environments.
> > >
> > > While we agree that this could be an interesting research direction, we also want to stress that reformulating the problem this way would mean that the (newly targeted) gains of the RANa model would have a very different nature and origin: instead of providing additional information on the actual scene in which the agent navigates, they would augment the general reasoning capacity of the agent and re-use the training set in a different and complementary way, post-training.
> > >
> > > This could in principle be done, but it would require an entire re-design of the proposed approach. An important contribution of this paper is the use of a strong **geometric foundation model** (the DEBiT encoder) that can derive valuable pose hints from the retrieved images in the context set. This provides **directional information** and allows the agent to easily interpret the retrieved images as potential waypoint recommendations. This is strongly linked to the setting of the paper, in which the retrieval process provides information on the actual scene in which the agent navigates. It is far less clear whether this choice makes sense if the method has to change its strategy towards using the retrieval process to gain general navigation knowledge. In particular, in this case there would be no directional information to be exploited in the retrieved context images, thus the geometric foundation model would not be appropriate to represent context information.
> > >
> > > To summarize, exploring this direction requires a different context encoder, a different training setup and strategy, potentially new auxiliary losses, and would in essence lead to a different paper, as its current message would be diluted considerably. Therefore, we propose to defer this investigation to future work.

---

> > > > ### Comment · Reviewer_ufQa · 2025-09-17
> > > >
> > > > Thank you for the response and clarifications. I was thinking about how to make the proposed approach more generalizable, and believe this would be an interesting next step.

---

### Review · Reviewer_8vmt · 2025-08-28

**Summary Of Contributions:**

The paper presents a retrieval design for augmenting navigation agent. The agent is trained via RL to solve navigation tasks, the paper proposes to augment the agent's context information with database retrieval. The paper presents solid technical design and empirical results, showing the promise of the approach.

**Audience:**

Yes

**Audience Explanation:**

The paper presents findings in the utility of retrieval mechanism for RL training, which definitely can be of interest to certain TMLR readers.

**Claims And Evidence:**

Yes

**Claims Explanation:**

The paper's writing is relatively easy to follow. The paper lays out details on the technical designs of the retrieval systems, and how it gets integrated into a canonical RNN based navigation agent. The paper also carries out fairly extensive empirical study that showcases the gains the retrieval entails when trained in the navigation environment.

**Requested Changes:**

I have a few technical questions regarding the paper.

=== ***static vs. dynamic context*** ===

Examining results in Table 2, it seems that the static context brings about most of the gains while the dynamic context only improves slightly on top. Does it mean that for the sake of simplicity, it suffices to use static retrieval for augmented context, to gain most of the performance improvements? I wonder why dynamic context would not obtain further gains given the additional amount of flexibility it entails.

=== ***RL training curves*** ===

Since the paper is centered on training RL with augmented input via retrieval, I think it's valuable to put RL training curves into the main paper. All results in the paper show the final performance of the trained system, and hence does not reflect how augmented input helps with the performance. For example, do we expect speed improvement in learning? Do we expect better asymptotic performance during training? Are hyper-parameters more robust for the training curves due to retrieval?

=== ***Training retrieval process*** ===

I wonder if there is potential for training retrieval process itself, in addition to the encoder network that encodes the retrieved inputs? It seems that most of the prior work on this fixes the retrieval process, so does this current work. Just wonder if the authors have any thoughts regarding the challenges and potential with this approach.

=== ***Transformer architecture vs. rnn*** ===

While experiments with recurrent networks are certainly technical solid, as done in this work, I wonder whether the authors have considered the impact of retrieval on architectures, e.g., how it impacts the system of the base agent is transformer. An argument is that the recurrent architecture might be bottlenecked by the capacity of the hidden state, and hence retrieved inputs allow for improvements. However, when using transformer, the hidden state bottleneck is alleviated and I wonder if the gains entailed by retrieval will diminish as a result.

---

> ### Author Response · Authors · 2025-09-05
> **Response to the Review (1/2)**
>
> We thank the Reviewer for the extensive and constructive feedback. We are glad that the Reviewer found the paper easy to follow and of interest for TMLR readers, presenting a solid technical design, with fairly extensive empirical results showcasing the promise of the approach. In the following we address in detail the Requested Changes.
>
> > ***=== static vs. dynamic context ===***
> > Examining results in Table 2, it seems that the static context brings about most of the gains while the dynamic context only improves slightly on top. Does it mean that for the sake of simplicity, it suffices to use static retrieval for augmented context, to gain most of the performance improvements? I wonder why dynamic context would not obtain further gains given the additional amount of flexibility it entails.
>
> In the experimental results we present two metrics, Success Rate (SR) and Success weighted by Path Length (SPL), which measures path efficiency. Both metrics are important, and the dynamic context provides significant gains in terms of path efficiency, expressed by the SPL metric. This means that the agent takes less detours and reaches the goal faster. In particular, this is achieved in combination with the attention-based encoder because it can exploit all the sequence of elements in context path, as opposed to the Gumbel selector that picks only one context item.
>
> > ***=== RL training curves ===***
> >Since the paper is centered on training RL with augmented input via retrieval, I think it's valuable to put RL training curves into the main paper. All results in the paper show the final performance of the trained system, and hence does not reflect how augmented input helps with the performance. For example, do we expect speed improvement in learning? Do we expect better asymptotic performance during training? Are hyper-parameters more robust for the training curves due to retrieval?
>
> RANa agents are trained starting form DEBiT checkpoints, therefore training curves are stable and quickly converge to high values. In the revised manuscript we have included a paragraph and a new Figure 5, with training curves (Return and Success Rate) for RANa-t with *static* Gumbel context selector and DEBiT finetuned for additional 200M steps. Both models, RANa and the baseline, thus start from the same checkpoint for fairness. The behavior of the two models is comparable, but RANa achieves consistently higher values in both metrics.
>
> > ***=== Training retrieval process ===***
> > I wonder if there is potential for training retrieval process itself, in addition to the encoder network that encodes the retrieved inputs? It seems that most of the prior work on this fixes the retrieval process, so does this current work. Just wonder if the authors have any thoughts regarding the challenges and potential with this approach.
>
> Thanks for the excellent question, this is indeed one of the directions we are planning to investigate in future work.
>
> The retrieval process itself is non-differentiable, thus it is not possible to train the encoders used for retrieval end-to-end naively.
> One option is to extend  RANa with the approach proposed in *REALM: Retrieval Augmented Language Model Pre-Training (ICML 2020)*, where retrieval is formulated in a differentiable way and a few techniques are introduced to train the retrieval process end-to-end. While involving, the method showed promising results and has certainly the potential to improve RANa's performance.
>
> A potentially more manageable option could be to first train the retriever in a simpler proxy task in simulation using privileged information. For example, we could train the retrieval process with supervised learning to select the image (or images) from the database that best match the ones along the shortest path to the goal.

---

> > ### Author Response · Authors · 2025-09-05
> > **Response to the Review (2/2)**
> >
> > > ***=== Transformer architecture vs. rnn ===***
> > > While experiments with recurrent networks are certainly technical solid, as done in this work, I wonder whether the authors have considered the impact of retrieval on architectures, e.g., how it impacts the system of the base agent is transformer. An argument is that the recurrent architecture might be bottlenecked by the capacity of the hidden state, and hence retrieved inputs allow for improvements. However, when using transformer, the hidden state bottleneck is alleviated and I wonder if the gains entailed by retrieval will diminish as a result.
> >
> > This is a very pertinent remark.
> >
> > First, it is important to stress that not all RANa gains come from what can be modeled in the RNN or transformer memory, but it also comes from additional scene views stored in the database and never seen by the agent otherwise. These gains are arguably not bottlenecked by the memory capacity.
> >
> > Second, indeed RANa could fit very well with a transformer-based architecture, and we are actively exploring this direction. While it is true that the recurrent memory has a more limited capacity compared to transformers, a transformer-based agent comes with its own challenges. The computational complexity of attention over time grows quadratically with the length of the window attended, requiring to either use a short time window to limit latency at inference, which is sub-optimal to navigate in large scenes; or accept delays due to computational complexity, which is also problematic in real-world robotic applications.  We believe that RANa could alleviate these shortcomings by using a short time attention window processed by the transformer, and relying on the retrieved elements for more distant views of the environment.

---

### Author Response · Authors · 2025-09-05
**Overview of responses and manuscript changes**

We would like to thank the Action Editor and the Reviewers for their time and the valuable feedback that helped improve the paper. We are glad that the Reviewers found the paper easy to follow (**8vmt**, **mTWV**), proposing an interesting (**8vmt**, **ufQa**), smart (**ufQa**), powerful and scalable approach (**mTWV**), with a solid technical design (**8vmt**) that aligns well with modern deep learning paradigm where large-scale data can be converted into useful representations (**mTWV**), supported by extensive experiments (**8vmt**) demonstrating solid performance improvements (**8vmt**, **ufQa**) and impressive zero-shot performance (**ufQa**).

We have uploaded a revised manuscript to address the Reviewers' remarks, with the most significant changes highlighted in blue. The paper is updated in three main parts to address the remarks made by the Reviewers. The proposed modifications can be summarized as follows:
- **Clarification for “store” in Table 1**: On page 4, we have added a description of data storage for RANa in the caption of Table 1.
- **Discussion about failure cases and robustness**: On page 9, we have taken the content of the original  paragraph entitled *Context is important* and transformed and expanded  it into a new paragraph *Context robustness and failure cases*. This new paragraph now contains a discussion about robustness of RANa to misleading context elements, and also includes the failure case discussion previously presented in the Appendix.
- **RL training curves**: On page 10, we have included a paragraph and a new Figure 5 with training curves (Return and Success Rate) for RANa and the DEBiT baseline. The curves exhibit a similar behavior for the two models, but RANa achieves consistently higher values in both metrics.

---

### Decision · Action_Editor_MAHV · 2025-10-20

**Recommendation:** Accept as is

**Audience:**

Yes

**Audience Explanation:**

Researchers interested in navigation tasks, and more generally RL, should find this of interest.

**Claims And Evidence:**

Yes

**Claims Explanation:**

The authors provide a number of empirical evaluations supporting the efficacy of their proposed method. Based on reviewer feedback, the authors also included a discussion of failure cases, which helps provide a more accurate depiction of their approach.